# AUTOMATED RELATIONAL META-LEARNING

**Huaxiu Yao**[1]*, **Xian Wu**[2], **Zhiqiang Tao**[3], **Yaliang Li**[4], **Bolin Ding**[4], **Ruirui Li**[5], **Zhenhui Li**[1]
[1]Pennsylvania State University, [2]University of Notre Dame, [3]Northeastern University
[4]Alibaba Group, [5]University of California, Los Angeles
[1]{huaxiuyao,JessieLi}@psu.edu, [2]xwu9@nd.edu, [3]zqtao@ece.neu.edu
[4]{yaliang.li,bolin.ding}@alibaba-inc.com, [5]rrli@cs.ucla.edu

## ABSTRACT

In order to efficiently learn with small amount of data on new tasks, meta-learning transfers knowledge learned from previous tasks to the new ones. However, a critical challenge in meta-learning is the task heterogeneity which cannot be well handled by traditional globally shared meta-learning methods. In addition, current task-specific meta-learning methods may either suffer from hand-crafted structure design or lack the capability to capture complex relations between tasks. In this paper, motivated by the way of knowledge organization in knowledge bases, we propose an automated relational meta-learning (ARML) framework that automatically extracts the cross-task relations and constructs the meta-knowledge graph. When a new task arrives, it can quickly find the most relevant structure and tailor the learned structure knowledge to the meta-learner. As a result, the proposed framework not only addresses the challenge of task heterogeneity by a learned meta-knowledge graph, but also increases the model interpretability. We conduct extensive experiments on 2D toy regression and few-shot image classification and the results demonstrate the superiority of ARML over state-of-the-art baselines.

## 1 INTRODUCTION

Learning quickly is the key characteristic of human intelligence, which remains a daunting problem in machine intelligence. The mechanism of meta-learning is widely used to generalize and transfer prior knowledge learned from previous tasks to improve the effectiveness of learning on new tasks, which has benefited various applications, such as computer vision (Kang et al., 2019; Liu et al., 2019), natural language processing (Gu et al., 2018; Lin et al., 2019) and social good (Zhang et al., 2019; Yao et al., 2019a). Most of existing meta-learning algorithms learn a globally shared meta-learner (e.g., parameter initialization (Finn et al., 2017; 2018), meta-optimizer (Ravi & Larochelle, 2016), metric space (Snell et al., 2017; Garcia & Bruna, 2017; Oreshkin et al., 2018)). However, globally shared meta-learners fail to handle tasks lying in different distributions, which is known as *task heterogeneity* (Vuorio et al., 2018; Yao et al., 2019b). Task heterogeneity has been regarded as one of the most challenging issues in meta-learning, and thus it is desirable to design meta-learning models that effectively optimize each of the heterogeneous tasks.

The key challenge to deal with task heterogeneity is how to customize globally shared meta-learner by using task-specific information? Recently, a handful of works try to solve the problem by learning a task-specific representation for tailoring the transferred knowledge to each task (Oreshkin et al., 2018; Vuorio et al., 2018; Lee & Choi, 2018). However, the expressiveness of these methods is limited due to the impaired knowledge generalization between highly related tasks. Recently, learning the underlying structure across tasks provides a more effective way for balancing the customization and generalization. Representatively, Yao et al. propose a hierarchically structured meta-learning method to customize the globally shared knowledge to each cluster (Yao et al., 2019b). Nonetheless, the hierarchical clustering structure completely relies on the handcrafted design which needs to be tuned carefully and may lack the capability to capture complex relationships.

Hence, we are motivated to propose a framework to automatically extract underlying relational structures from historical tasks and leverage those relational structures to facilitate knowledge

---

*Work was done when Huaxiu Yao, Xian Wu, Zhiqiang Tao interned in Alibaba Group.

customization on a new task. This inspiration comes from the way of structuring knowledge in knowledge bases (i.e., knowledge graphs). In knowledge bases, the underlying relational structures across text entities are automatically constructed and applied to a new query to improve the searching efficiency. In the meta-learning problem, similarly, we aim at automatically establishing the meta-knowledge graph between prior knowledge learned from previous tasks. When a new task arrives, it queries the meta-knowledge graph and quickly attends to the most relevant entities (vertices), and then takes advantage of the relational knowledge structures between them to boost the learning effectiveness with the limited training data.

The proposed meta-learning framework is named as **A**utomated **R**elational **M**eta-**L**earning (ARML). Specifically, the ARML automatically builds the meta-knowledge graph from meta-training tasks to memorize and organize learned knowledge from historical tasks, where each vertex represents one type of meta-knowledge (e.g., the common contour between birds and aircrafts). To learn the meta-knowledge graph at meta-training time, for each task, we construct a prototype-based relational graph for each class, where each vertex represents one prototype. The prototype-based relational graph not only captures the underlying relationship behind samples, but alleviates the potential effects of abnormal samples. The meta-knowledge graph is then learned by summarizing the information from the corresponding prototype-based relational graphs of meta-training tasks. After constructing the meta-knowledge graph, when a new task comes in, the prototype-based relational graph of the new task taps into the meta-knowledge graph for acquiring the most relevant knowledge, which further enhances the task representation and facilitates its training process.

Our major contributions of the proposed ARML are three-fold: (1) it automatically constructs the meta-knowledge graph to facilitate learning a new task; (2) it empirically outperforms the state-of-the-art meta-learning algorithms; (3) the meta-knowledge graph well captures the relationship among tasks and improves the interpretability of meta-learning algorithms.

## 2 RELATED WORK

Meta-learning designs models to learn new tasks or adapt to new environments quickly with a few training examples. There are mainly three research lines of meta-learning: (1) black-box amortized methods design black-box meta-learners to infer the model parameters (Ravi & Larochelle, 2016; Andrychowicz et al., 2016; Mishra et al., 2018; Gordon et al., 2019); (2) gradient-based methods aim to learn an optimized initialization of model parameters, which can be adapted to new tasks by a few steps of gradient descent (Finn et al., 2017; 2018; Lee & Choi, 2018; Yoon et al., 2018; Grant et al., 2018); (3) non-parametric methods combine parametric meta-learners and non-parametric learners to learn an appropriate distance metric for few-shot classification (Snell et al., 2017; Vinyals et al., 2016; Yang et al., 2018; Oreshkin et al., 2018; Yoon et al., 2019; Garcia & Bruna, 2017).

Our work is built upon the gradient-based meta-learning methods. In the line of gradient-based meta-learning, most algorithms learn a globally shared meta-learners from previous tasks (Finn et al., 2017; Li et al., 2017; Flennerhag et al., 2019), to improve the effectiveness of learning process on new tasks. However, these algorithms typically lack the ability to handle heterogeneous tasks (i.e., tasks sample from sufficient different distributions). To tackle this challenge, recent works tailor the globally shared initialization to different tasks by customizing initialization (Vuorio et al., 2018; Yao et al., 2019b) and using probabilistic models (Yoon et al., 2018; Finn et al., 2018). Representatively, HSML customizes the globally shared initialization with a manually designed hierarchical clustering structure to balance the generalization and customization (Yao et al., 2019b). However, the hand-crafted designed hierarchical structure may not accurately reflect the real structure and the clustering structure constricts the complexity of relationship. Compared with these methods, ARML leverages the most relevant structure from the automatically constructed meta-knowledge graph. Thus, ARML not only discovers more accurate underlying structures to improve the effectiveness of meta-learning algorithms, but also the meta-knowledge graph further enhances the model interpretability.

## 3 PRELIMINARIES

**Few-shot Learning**     Considering a task $\mathcal{T}_i$, the goal of few-shot learning is to learn a model with a dataset $\mathcal{D}_i = \{\mathcal{D}_i^{tr}, \mathcal{D}_i^{ts}\}$, where the labeled training set $\mathcal{D}_i^{tr} = \{\mathbf{x}_j^{tr}, \mathbf{y}_j^{tr} | \forall j \in [1, N^{tr}]\}$ only has a few samples and $\mathcal{D}_i^{ts}$ represents the corresponding test set. A learning model (a.k.a., base model) $f$

with parameters $\theta$ are used to evaluate the effectiveness on $\mathcal{D}_i^{ts}$ by minimizing the expected empirical loss on $\mathcal{D}_i^{tr}$, i.e., $\mathcal{L}(\mathcal{D}_{\mathcal{T}_i}^{tr}, \theta)$, and obtain the optimal parameters $\theta_i$. For the regression problem, the loss function is defined based on the mean square error (i.e., $\sum_{(\mathbf{x}_j, \mathbf{y}_j) \in \mathcal{D}_i^{tr}} \|f_\theta(\mathbf{x}_j) - \mathbf{y}_j\|_2^2$) and for the classification problem, the loss function uses the cross-entropy loss (i.e., $-\sum_{(\mathbf{x}_j, \mathbf{y}_j) \in \mathcal{D}_i^{tr}} \log p(\mathbf{y}_j | \mathbf{x}_j, f_\theta)$). Usually, optimizing and learning parameter $\theta$ for the task $\mathcal{T}_i$ with a few labeled training samples is difficult. To address this limitation, meta-learning provides us a new perspective to improve the performance by leveraging knowledge from multiple tasks.

**Meta-learning and Model-agnostic Meta-learning**  In meta-learning, a sequence of tasks $\{\mathcal{T}_1, ..., \mathcal{T}_I\}$ are sampled from a task-level probability distribution $p(\mathcal{T})$, where each one is a few-shot learning task. To facilitate the adaption for incoming tasks, the meta-learning algorithm aims to find a well-generalized meta-learner on $I$ training tasks at meta-learning phase. At meta-testing phase, the optimal meta-learner is applied to adapt the new tasks $\mathcal{T}_t$. In this way, meta-learning algorithms are capable of adapting to new tasks efficiently even with a shortage of training data for a new task.

Model-agnostic meta-learning (MAML) (Finn et al., 2017), one of the representative algorithms in gradient-based meta-learning, regards the meta-learner as the initialization of parameter $\theta$, i.e., $\theta_0$, and learns a well-generalized initialization $\theta_0^*$ during the meta-training process. The optimization problem is formulated as (one gradient step as exemplary):

$$\theta_0^* := \arg\min_{\theta_0} \sum_{i=1}^{I} \mathcal{L}(f_{\theta_i}, \mathcal{D}_i^{ts}) = \arg\min_{\theta_0} \sum_{i=1}^{I} \mathcal{L}(f_{\theta_0 - \alpha \nabla_\theta \mathcal{L}(f_\theta, \mathcal{D}_i^{tr})}, \mathcal{D}_i^{ts}). \tag{1}$$

At the meta-testing phase, to obtain the adaptive parameter $\theta_t$ for each new task $\mathcal{T}_t$, we finetune the initialization of parameter $\theta_0^*$ by performing gradient updates a few steps, i.e., $f_{\theta_t} = f_{\theta_0^* - \alpha \nabla_\theta \mathcal{L}(f_\theta, \mathcal{D}_t^{tr})}$.

# 4 METHODOLOGY

In this section, we introduce the details of the proposed ARML. To better explain how it works, we show its framework in Figure 1. The goal of ARML is to facilitate the learning process of new tasks by leveraging transferable knowledge learned from historical tasks. To achieve this goal, we introduce a meta-knowledge graph, which is automatically constructed at the meta-training time, to organize and memorize historical learned knowledge. Given a task, which is built as a prototype-based relational structure, it taps into the meta-knowledge graph to acquire relevant knowledge for enhancing its own representation. The enhanced prototype representations further aggregate and incorporate with meta-learner for fast and effective adaptions by utilizing a modulating function. In the following subsections, we elaborate three key components: prototype-based sample structuring, automated meta-knowledge graph construction and utilization, and task-specific knowledge fusion and adaptation, respectively.

## 4.1 PROTOTYPE-BASED SAMPLE STRUCTURING

Given a task which involves either classifications or regressions regarding a set of samples, we first investigate the relationships among these samples. Such relationship is represented by a graph, called prototype-based relational graph in this work, where the vertices in the graph denote the prototypes of different classes while the edges and the corresponding edge weights are created based on the similarities between prototypes. Constructing the relational graph based on prototypes instead of raw samples allows us to alleviate the issue raised by abnormal samples. As the abnormal samples, which locate far away from normal samples, could pose significant concerns especially when only a limited number of samples are available for training. Specifically, for classification problem, the prototype, denoted by $\mathbf{c}_i^k \in \mathbb{R}^d$, is defined as:

$$\mathbf{c}_i^k = \frac{1}{N_k^{tr}} \sum_{j=1}^{N_k^{tr}} \mathcal{E}(\mathbf{x}_j), \tag{2}$$

where $N_k^{tr}$ denotes the number of samples in class $k$. $\mathcal{E}$ is an embedding function, which projects $\mathbf{x}_j$ into a hidden space where samples from the same class are located closer to each other while samples from different classes stay apart. For regression problem, it is not straightforward to construct

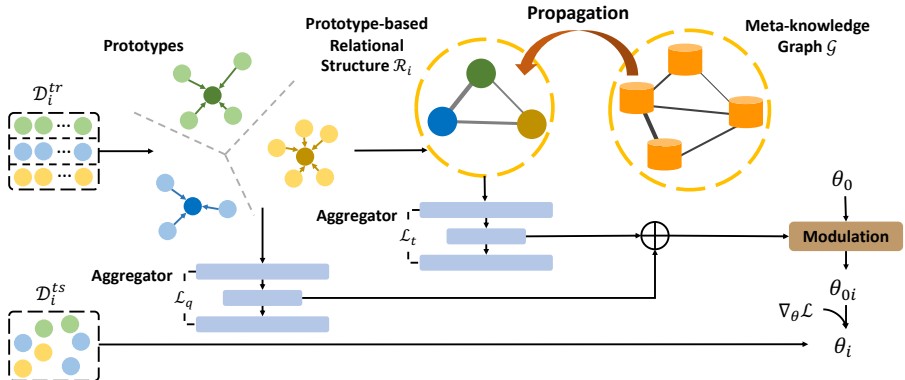

Figure 1: The framework of ARML. For each task $\mathcal{T}_i$, ARML first builds a prototype-based relational structure $\mathcal{R}_i$ by mapping the training samples $\mathcal{D}_i^{tr}$ into prototypes, with each prototype represents one class. Then, $\mathcal{R}_i$ interacts with the meta-knowledge graph $\mathcal{G}$ to acquire the most relevant historical knowledge by information propagation. Finally, the task-specific modulation tailors the globally shared initialization $\theta_0$ by aggregating of raw prototypes and enriched prototypes, which absorbs relevant historical information from the meta-knowledge graph.

the prototypes explicitly based on class information. Therefore, we cluster samples by learning an assignment matrix $\mathbf{P}_i \in \mathbb{R}^{K \times N^{tr}}$. Specifically, we formulate the process as:

$$\mathbf{P}_i = \text{Softmax}(\mathbf{W}_p \mathcal{E}^{\mathrm{T}}(\mathbf{X}) + \mathbf{b}_p), \ \mathbf{c}_i^k = \mathbf{P}_i[k]\mathcal{F}(\mathbf{X}), \tag{3}$$

where $\mathbf{P}_i[k]$ represents the $k$-th row of $\mathbf{P}_i$. Thus, training samples are clustered to $K$ clusters, which serve as the representation of prototypes.

After calculating all prototype representations $\{\mathbf{c}_i^k | \forall k \in [1, K]\}$, which serve as the vertices in the the prototype-based relational graph $\mathcal{R}_i$, we further define the edges and the corresponding edge weights. The edge weight $A_{\mathcal{R}_i}(\mathbf{c}_i^j, \mathbf{c}_i^m)$ between two prototypes $\mathbf{c}_i^j$ and $\mathbf{c}_i^m$ is gauged by the the similarity between them. Formally:

$$A_{\mathcal{R}_i}(\mathbf{c}_i^j, \mathbf{c}_i^m) = \sigma(\mathbf{W}_r(|\mathbf{c}_i^j - \mathbf{c}_i^m|/\gamma_r) + \mathbf{b}_r), \tag{4}$$

where $\mathbf{W}_r$ and $\mathbf{b}_r$ represents learnable parameters, $\gamma_r$ is a scalar and $\sigma$ is the Sigmoid function, which normalizes the weight between 0 and 1. For simplicity, we denote the prototype-based relational graph as $\mathcal{R}_i = (\mathbf{C}_{\mathcal{R}_i}, \mathbf{A}_{\mathcal{R}_i})$, where $\mathbf{C}_{\mathcal{R}_i} = \{\mathbf{c}_i^j | \forall j \in [1, K]\} \in \mathbb{R}^{K \times d}$ represent a set of vertices, with each one corresponds to the prototype from a class, while $\mathbf{A}_{\mathcal{R}_i} = \{|A_{\mathcal{R}_i}(\mathbf{c}_i^j, \mathbf{c}_i^m)| \forall j, m \in [1, K]\} \in \mathbb{R}^{K \times K}$ gives the adjacency matrix, which indicates the proximity between prototypes.

## 4.2 Automated Meta-knowledge Graph Construction and Utilization

In this section, we first discuss how to organize and distill knowledge from historical learning process and then expound how to leverage such knowledge to benefit the training of new tasks. To organize and distill knowledge from historical learning process, we construct and maintain a meta-knowledge graph. The vertices represent different types of meta-knowledge (e.g., the common contour between aircrafts and birds) and the edges are automatically constructed to reflect the relationship between meta-knowledge. When serving a new task, we refer to the meta-knowledge, which allows us to efficiently and automatically identify relational knowledge from previous tasks. In this way, the training of a new task can benefit from related training experience and get optimized much faster than otherwise possible. In this paper, the meta-knowledge graph is automatically constructed at the meta-training phase. The details of the construction are elaborated as follows:

Assuming the representation of an vertex $g$ is given by $\mathbf{h}^g \in \mathbb{R}^d$, we define the meta-knowledge graph as $\mathcal{G} = (\mathbf{H}_{\mathcal{G}}, \mathbf{A}_{\mathcal{G}})$, where $\mathbf{H}_{\mathcal{G}} = \{\mathbf{h}^j | \forall j \in [1, G]\} \in \mathbb{R}^{G \times d}$ and $\mathbf{A}_{\mathcal{G}} = \{A_{\mathcal{G}}(\mathbf{h}^j, \mathbf{h}^m) | \forall j, m \in [1, G]\} \in \mathbb{R}^{G \times G}$ denote the vertex feature matrix and vertex adjacency matrix, respectively. To better explain the construction of the meta-knowledge graph, we first discuss the vertex representation $\mathbf{H}_{\mathcal{G}}$. During meta-training, tasks arrive one after another in a sequence and their corresponding vertices

representations are expected to be updated dynamically in a timely manner. Therefore, the vertex representation of meta-knowledge graph are defined to get parameterized and learned at the training time. Moreover, to encourage the diversity of meta-knowledge encoded in the meta-knowledge graph, the vertex representations are randomly initialized. Analogous to the definition of weight in the prototype-based relational graph $\mathcal{R}_i$ in equation 4, the weight between a pair of vertices $j$ and $m$ is constructed as:

$$A_{\mathcal{G}}(\mathbf{h}^j, \mathbf{h}^m) = \sigma(\mathbf{W}_o(|\mathbf{h}^j - \mathbf{h}^m|/\gamma_o) + \mathbf{b}_o), \tag{5}$$

where $\mathbf{W}_o$ and $\mathbf{b}_o$ represent learnable parameters and $\gamma_o$ is a scalar.

To enhance the learning of new tasks with involvement of historical knowledge, we query the prototype-based relational graph in the meta-knowledge graph to obtain the relevant knowledge in history. The ideal query mechanism is expected to optimize both graph representations simultaneously at the meta-training time, with the training of one graph facilitating the training of the other. In light of this, we construct a super-graph $\mathcal{S}_i$ by connecting the prototype-based relational graph $\mathcal{R}_i$ with the meta-knowledge graph $\mathcal{G}$ for each task $\mathcal{T}_i$. The union of the vertices in $\mathcal{R}_i$ and $\mathcal{G}$ contributes to the vertices in the super-graph. The edges in $\mathcal{R}_i$ and $\mathcal{G}$ are also reserved in the super-graph. We connect $\mathcal{R}_i$ with $\mathcal{G}$ by creating links between the prototype-based relational graph with the meta-knowledge graph. The link between prototype $\mathbf{c}_i^j$ in prototype-based relational graph and vertex $\mathbf{h}^m$ in meta-knowledge graph is weighted by the similarity between them. More precisely, for each prototype $\mathbf{c}_i^j$, the link weight $A_{\mathcal{S}}(\mathbf{c}_i^j, \mathbf{h}^m)$ is calculated by applying softmax over Euclidean distances between $\mathbf{c}_i^j$ and $\{\mathbf{h}^m | \forall m \in [1, G]\}$ as follows:

$$A_{\mathcal{S}}(\mathbf{c}_i^j, \mathbf{h}^k) = \frac{\exp(-\|(\mathbf{c}_i^j - \mathbf{h}^k)/\gamma_s\|_2^2/2)}{\sum_{k'=1}^{K} \exp(-\|(\mathbf{c}_i^j - \mathbf{h}^{k'})/\gamma_s\|_2^2/2)}, \tag{6}$$

where $\gamma_s$ is a scaling factor. We denote the intra-adjacent matrix as $\mathbf{A}_{\mathcal{S}} = \{A_{\mathcal{S}}(\mathbf{c}_i^j, \mathbf{h}^m) | \forall j \in [1, K], m \in [1, G]\} \in \mathbb{R}^{K \times G}$. Thus, for task $\mathcal{T}_i$, the adjacent matrix and feature matrix of super-graph $\mathcal{S}_i = (\mathbf{A}_i, \mathbf{H}_i)$ is defined as $\mathbf{A}_i = (\mathbf{A}_{\mathcal{R}_i}, \mathbf{A}_{\mathcal{S}}; \mathbf{A}_{\mathcal{S}}^{\mathrm{T}}, \mathbf{A}_{\mathcal{G}}) \in \mathbb{R}^{(K+G) \times (K+G)}$ and $\mathbf{H}_i = (\mathbf{C}_{\mathcal{R}_i}; \mathbf{H}_{\mathcal{G}}) \in \mathbb{R}^{(K+G) \times d}$, respectively.

After constructing the super-graph $\mathcal{S}_i$, we are able to propagate the most relevant knowledge from meta-knowledge graph $\mathcal{G}$ to the prototype-based relational graph $\mathcal{R}_i$ by introducing a Graph Neural Networks (GNN). In this work, following the "message-passing" framework (Gilmer et al., 2017), the GNN is formulated as:

$$\mathbf{H}_i^{(l+1)} = \mathrm{MP}(\mathbf{A}_i, \mathbf{H}_i^{(l)}; \mathbf{W}^{(l)}), \tag{7}$$

where $\mathrm{MP}(\cdot)$ is the message passing function and has several possible implementations (Hamilton et al., 2017; Kipf & Welling, 2017; Veličković et al., 2018), $\mathbf{H}_i^{(l)}$ is the vertex embedding after $l$ layers of GNN and $\mathbf{W}^{(l)}$ is a learnable weight matrix of layer $l$. The input $\mathbf{H}_i^{(0)} = \mathbf{H}_i$. After stacking $L$ GNN layers, we get the information-propagated feature representation for the prototype-based relational graph $\mathcal{R}_i$ as the top-$K$ rows of $\mathbf{H}_i^{(L)}$, which is denoted as $\hat{\mathbf{C}}_{\mathcal{R}_i} = \{\hat{\mathbf{c}}_i^j | j \in [1, K]\}$.

### 4.3 Task-specific Knowledge Fusion and Adaptation

After propagating information form meta-knowledge graph to prototype-based relational graph, in this section, we discuss how to learn a well-generalized meta-learner for fast and effective adaptions to new tasks with limited training data. To tackle the challenge of task heterogeneity, in this paper, we incorporate task-specific information to customize the globally shared meta-learner (e.g., initialization here) by leveraging a modulating function, which has been proven to be effective to provide customized initialization in previous studies (Wang et al., 2019; Vuorio et al., 2018).

The modulating function relies on well-discriminated task representations, while it is difficult to learn all representations by merely utilizing the loss signal derived from the test set $\mathcal{D}_i^{ts}$. To encourage such stability, we introduce two reconstructions by utilizing two auto-encoders. There are two collections of parameters, i.e, $\mathbf{C}_{\mathcal{R}_i}$ and $\hat{\mathbf{C}}_{\mathcal{R}_i}$, which contribute the most to the creation of the task-specific meta-learner. $\mathbf{C}_{\mathcal{R}_i}$ express the raw prototype information without tapping into the meta-knowledge graph, while $\hat{\mathbf{C}}_{\mathcal{R}_i}$ give the prototype representations after absorbing the relevant knowledge from the meta-knowledge graph. Therefore, the two reconstructions are built on $\mathbf{C}_{\mathcal{R}_i}$ and $\hat{\mathbf{C}}_{\mathcal{R}_i}$. To reconstruct $\mathbf{C}_{\mathcal{R}_i}$, an aggregator $\mathrm{AG}^q(\cdot)$ (e.g., recurrent network, fully connected layers) is involved to encode $\mathbf{C}_{\mathcal{R}_i}$ into a dense representation, which is further fed into a decoder $\mathrm{AG}_{dec}^q(\cdot)$ to achieve reconstructions.

---

**Algorithm 1** Meta-Training Process of ARML

---

**Require:** $p(\mathcal{T})$: distribution over tasks; $K$: Number of vertices in meta-knowledge graph; $\alpha$: stepsize for gradient descent of each task (i.e., inner loop stepsize); $\beta$: stepsize for meta-optimization (i.e., outer loop stepsize); $\mu_1, \mu_2$: balancing factors in loss function

1: Randomly initialize all learnable parameters $\Phi$
2: **while** not done **do**
3:     Sample a batch of tasks $\{\mathcal{T}_i | i \in [1, I]\}$ from $p(\mathcal{T})$
4:     **for all** $\mathcal{T}_i$ **do**
5:         Sample training set $\mathcal{D}_i^{tr}$ and testing set $\mathcal{D}_i^{ts}$
6:         Construct the prototype-based relational graph $\mathcal{R}_i$ by computing prototype in equation 2 and weight in equation 4
7:         Compute the similarity between each prototype and meta-knowledge vertex in equation 6 and construct the super-graph $\mathcal{S}_i$
8:         Apply GNN on super-graph $\mathcal{S}_i$ and get the information-propagated representation $\hat{\mathbf{C}}_{\mathcal{R}_i}$
9:         Aggregate $\mathbf{C}_{\mathcal{R}_i}$ in equation 8 and $\hat{\mathbf{C}}_{\mathcal{R}_i}$ in equation 9 to get the representations $\mathbf{q}_i, \mathbf{t}_i$ and reconstruction loss $\mathcal{L}_q, \mathcal{L}_t$
10:        Compute the task-specific initialization $\theta_{0i}$ in equation 10 and update parameters $\theta_i = \theta_{0i} - \alpha \nabla_\theta \mathcal{L}(f_\theta, \mathcal{D}_i^{tr})$
11:     **end for**
12:     Update $\Phi \leftarrow \Phi - \beta \nabla_\Phi \sum_{i=1}^{I} \mathcal{L}(f_{\theta_i}, \mathcal{D}_i^{ts}) + \mu_i \mathcal{L}_t + \mu_2 \mathcal{L}_q$
13: **end while**

---

Then, the corresponded task representation $\mathbf{q}_i$ of $\mathbf{C}_{\mathcal{R}_i}$ is summarized by applying a mean pooling operator over prototypes on the encoded dense representation. Formally,

$$\mathbf{q}_i = \text{MeanPool}(\text{AG}^q(\mathbf{C}_{\mathcal{R}_i})) = \frac{1}{N^{tr}} \sum_{j=1}^{N^{tr}} (\text{AG}^q(\mathbf{c}_i^j)), \quad \mathcal{L}_q = \|\mathbf{C}_{\mathcal{R}_i} - \text{AG}_{dec}^q(\text{AG}^q(\mathbf{C}_{\mathcal{R}_i}))\|_F^2 \quad (8)$$

Similarly, we reconstruct $\hat{\mathbf{C}}_{\mathcal{R}_i}$ and get the corresponded task representation $\mathbf{t}_i$ as follows:

$$\mathbf{t}_i = \text{MeanPool}(\text{AG}^t(\hat{\mathbf{C}}_{\mathcal{R}_i})) = \frac{1}{N^{tr}} \sum_{j=1}^{N^{tr}} (\text{AG}^t(\hat{\mathbf{c}}_i^j)), \quad \mathcal{L}_t = \|\hat{\mathbf{C}}_{\mathcal{R}_i} - \text{AG}_{dec}^t(\text{AG}^t(\hat{\mathbf{C}}_{\mathcal{R}_i}))\|_F^2 \quad (9)$$

The reconstruction errors in Equations 8 and 9 pose an extra constraint to enhance the training stability, leading to improvement of task representation learning.

After getting the task representation $\mathbf{q}_i$ and $\mathbf{t}_i$, the modulating function is then used to tailor the task-specific information to the globally shared initialization $\theta_0$, which is formulated as:

$$\theta_{0i} = \sigma(\mathbf{W}_g(\mathbf{t}_i \oplus \mathbf{q}_i) + \mathbf{b}_g) \circ \theta_0, \quad (10)$$

where $\mathbf{W}_g$ and $\mathbf{b}_g$ is learnable parameters of a fully connected layer. Note that we adopt the Sigmoid gating as exemplary and more discussion about different modulating functions can be found in ablation studies of Section 5.

For each task $\mathcal{T}_i$, we perform the gradient descent process from $\theta_{0i}$ and reach its optimal parameter $\theta_i$. Combining the reconstruction loss $\mathcal{L}_t$ and $\mathcal{L}_q$ with the meta-learning loss defined in equation 1, the overall objective function of ARML is:

$$\min_\Phi \mathcal{L}_{all} = \min_\Phi \mathcal{L} + \mu_1 \mathcal{L}_t + \mu_2 \mathcal{L}_q = \min_\Phi \sum_{i=1}^{I} \mathcal{L}(f_{\theta_{0i} - \alpha \nabla_\theta \mathcal{L}(f_\theta, \mathcal{D}_i^{tr})}, \mathcal{D}_i^{ts}) + \mu_1 \mathcal{L}_t + \mu_2 \mathcal{L}_q, \quad (11)$$

where $\mu_1$ and $\mu_2$ are introduced to balance the importance of these three items. $\Phi$ represents all learnable parameters. The algorithm of meta-training process of ARML is shown in Alg. 2. The details of the meta-testing process of ARML are available in Appendix A.

## 5 EXPERIMENTS

In this section, we conduct extensive experiments to demonstrate the effectiveness of the ARML on 2D regression and few-shot classification.

## 5.1 EXPERIMENTAL SETTINGS

**Methods for Comparison**  We compare our proposed ARML with two types of baselines: (1) *Gradient-based meta-learning methods*: both globally shared methods (MAML (Finn et al., 2017), Meta-SGD (Li et al., 2017)) and task-specific methods (MT-Net (Lee & Choi, 2018), MUMO-MAML (Vuorio et al., 2018), HSML (Yao et al., 2019b), BMAML (Yoon et al., 2018)) are considered for comparison. (2) *Other meta-learning methods (non-parametric and black box amortized methods)*: we select globally shared methods VERSA (Gordon et al., 2019), Prototypical Network (ProtoNet) (Snell et al., 2017), TapNet (Yoon et al., 2019) and task-specific method TADAM (Oreshkin et al., 2018) as baselines. Following the traditional settings, non-parametric baselines are only used in few-shot classification. Detailed implementations of baselines are discussed in Appendix B.3.

**Hyperparameter Settings**  For the aggregated function in autoencoder structure ($AG^t$, $AG^t_{dec}$ $AG^q$, $AG^q_{dec}$), we use the GRU as the encoder and decoder in this structure. We adopt one layer GCN (Kipf & Welling, 2017) with tanh activation as the implementation of GNN in equation 7. For the modulation network, we test sigmoid, tanh and Film modulation, and find that sigmoid modulation achieves best performance. Thus, in the future experiment, we set the sigmoid modulation as modulating function. More detailed discussion about experiment settings are presented in Appendix B.

## 5.2 2D REGRESSION

**Dataset Description**  In 2D regression problem, we adopt the similar regression problem settings as (Finn et al., 2018; Vuorio et al., 2018; Yao et al., 2019b; Rusu et al., 2019), which includes several families of functions. In this paper, to model more complex relational structures, we design a 2D regression problem rather than traditional 1D regression. Input $x \sim U[0.0, 5.0]$ and $y \sim U[0.0, 5.0]$ are sampled randomly and random Gaussian noisy with standard deviation 0.3 is added to the output. Furthermore, six underlying functions are selected, including (1) *Sinusoids:* $z(x, y) = a_s sin(w_s x + b_s)$, where $a_s \sim U[0.1, 5.0]$, $b_s \sim U[0, 2\pi]$ $w_s \sim U[0.8, 1.2]$; (2) *Line:* $z(x, y) = a_l x + b_l$, where $a_l \sim U[-3.0, 3.0]$, $b_l \sim U[-3.0, 3.0]$; (3) *Quadratic:* $z(x, y) = a_q x^2 + b_q x + c_q$, where $a_q \sim U[-0.2, 0.2]$, $b_q \sim U[-2.0, 2.0]$, $c_q \sim U[-3.0, 3.0]$; (4) *Cubic:* $z(x, y) = a_c x^3 + b_c x^2 + c_c x + d_c$, where $a_c \sim U[-0.1, 0.1]$, $b_c \sim U[-0.2, 0.2]$, $c_c \sim U[-2.0, 2.0]$, $d_c \sim U[-3.0, 3.0]$; (5) *Quadratic Surface:* $z(x, y) = a_{qs} x^2 + b_{qs} y^2$, where $a_{qs} \sim U[-1.0, 1.0]$, $b_{qs} \sim U[-1.0, 1.0]$; (6) *Ripple:* $z(x, y) = sin(-a_r(x^2 + y^2)) + b_r$, where $a_r \sim U[-0.2, 0.2]$, $b_r \sim U[-3.0, 3.0]$. Note that, function 1-4 are located in the subspace of $y = 1$. Follow (Finn et al., 2017), we use two fully connected layers with 40 neurons as the base model. The number of vertices of meta-knowledge graph is set as 6.

**Results and Analysis**  In Figure 2, we summarize the interpretation of meta-knowledge graph (see top figure, and more cases are provided in Figure 8 of Appendix G.4) and the the qualitative results (see bottom table) of 10-shot 2D regression. In the bottom table, we can observe that ARML achieves the best performance as compared to competitive gradient-based meta-learning methods, i.e., globally shared models and task-specific models. This finding demonstrates that the meta-knowledge graph is necessary to model and capture task-specific information. The superior performance can also be interpreted in the top figure. In the left, we show the heatmap between prototypes and meta-knowledge vertices (darker color means higher similarity). We can see that sinusoids and line activate V1 and V4, which may represent curve and line, respectively. V1 and V4 also contribute to quadratic and quadratic surface, which also show the similarity between these two families of functions. V3 is activated in P0 of all functions and the quadratic surface and ripple further activate V1 in P0, which may show the different between 2D functions and 3D functions (sinusoid, line, quadratic and cubic lie in the subspace). Specifically, in the right figure, we illustrate the meta-knowledge graph, where we set a threshold to filter the link with low similarity score and show the rest. We can see that V3 is the most popular vertice and connected with V1, V5 (represent curve) and V4 (represent line). V1 is further connected with V5, demonstrating the similarity of curve representation.

## 5.3 FEW-SHOT CLASSIFICATION

**Dataset Description and Settings**  In the few-shot classification problem, we first use the benchmark proposed in (Yao et al., 2019b), where four fine-grained image classification datasets are included (i.e., CUB-200-2011 (Bird), Describable Textures Dataset (Texture), FGVC of Aircraft

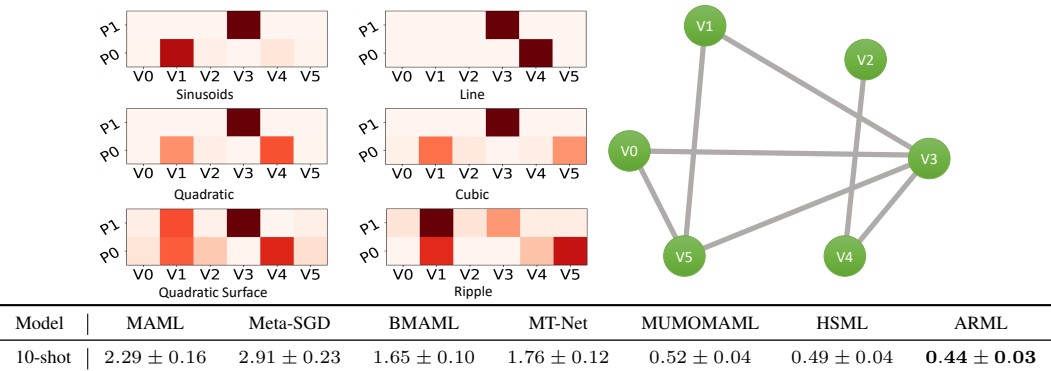

| Model | MAML | Meta-SGD | BMAML | MT-Net | MUMOMAML | HSML | ARML |
|---|---|---|---|---|---|---|---|
| 10-shot | $2.29 \pm 0.16$ | $2.91 \pm 0.23$ | $1.65 \pm 0.10$ | $1.76 \pm 0.12$ | $0.52 \pm 0.04$ | $0.49 \pm 0.04$ | $\mathbf{0.44 \pm 0.03}$ |

Figure 2: In the top figure, we show the interpretation of meta-knowledge graph. The left heatmap shows the similarity between prototypes (P0, P1) and meta-knowledge vertices (V0-V5). The right part show the meta-knowledge graph. In the bottom table, we show the overall performance (mean square error with 95% confidence) of 10-shot 2D regression.

(Aircraft), and FGVCx-Fungi (Fungi)). For each few-shot classification task, it samples classes from one of four datasets. In this paper, we call this dataset as *Plain-Multi* and each fine-grained dataset as subdataset.

Then, to demonstrate the effectiveness of our proposed model for handling more complex underlying structures, in this paper, we increase the difficulty of few-shot classification problem by introducing two image filters: blur filter and pencil filter. Similar to (Jerfel et al., 2019), for each image in Plain-Multi, one artistic filters are applied to simulate a changing distribution of few-shot classification tasks. After applying the filters, the total number of subdatasets is 12 and each task is sampled from one of them. This data is named as *Art-Multi*. More detailed descriptions of the effect of different filters are discussed in Appendix C.

Following the traditional meta-learning settings, all datasets are divided into meta-training, meta-validation and meta-testing classes. The traditional N-way K-shot settings are used to split training and test set for each task. We adopt the standard four-block convolutional layers as the base learner (Finn et al., 2017; Snell et al., 2017) for ARML and all baselines for a fair comparison. The number of vertices of meta-knowledge graph for Plain-Multi and Art-Multi datasets are set as 4 and 8, respectively. Additionally, for the miniImagenet and tieredImagenet (Ren et al., 2018), similar to (Finn et al., 2018), which tasks are constructed from a single domain and do not have heterogeneity, we compare our proposed ARML with baseline models and present the results in Appendix D.

**Overall Performance** Experimental results for Plain-Multi and Art-Multi are shown in Table 1 and Table 2, respectively. For each dataset, the performance accuracy with 95% confidence interval is reported. Due to the space limitation, in Art-Multi dataset, we only show the average value of each filter here. The full results are shown in Table 8 of Appendix E. In these two tables, first, we can observe that task-specific gradient-based models (MT-Net, MUMOMAML, HSML, BMAML) significantly outperforms globally shared models (MAML, Meta-SGD). Second, compared ARML with other task-specific gradient-based meta-learning methods, the better performance confirms that ARML can model and extract task-specific information more accurately by leveraging the constructed meta-knowledge graph. Especially, the performance gap between the ARML and HSML verifies the benefits of relational structure compared with hierarchical clustering structure. Third, as a gradient-based meta-learning algorithm, ARML can also outperform methods of other research lines (i.e., ProtoNet, TADAM, TapNet and VERSA). Finally, to show the effectiveness of proposed components in ARML, we conduct comprehensive ablation studies in Appendix F. The results further demonstrate the effectiveness of prototype-based relational graph and meta-knowledge graph.

**Analysis of Constructed Meta-knowledge Graph** In this section, we conduct extensive qualitative analysis for the constructed meta-knowledge graph, which is regarded as the key component in ARML. Due to the space limit, we present the results on Art-Multi datasets here and the analysis of Plain-Multi with similar observations is discussed in Appendix G.1. We further analyze the effect of

Table 1: Overall few-shot classification results (accuracy $\pm$ 95% confidence) on Plain-Multi dataset.

| Settings | Algorithms | Data: Bird | Data: Texture | Data: Aircraft | Data: Fungi |
|---|---|---|---|---|---|
| 5-way 1-shot | VERSA | $53.40 \pm 1.41\%$ | $30.43 \pm 1.30\%$ | $50.60 \pm 1.34\%$ | $40.40 \pm 1.40\%$ |
| | ProtoNet | $54.11 \pm 1.38\%$ | $32.52 \pm 1.28\%$ | $50.63 \pm 1.35\%$ | $41.05 \pm 1.37\%$ |
| | TapNet | $54.90 \pm 1.34\%$ | $32.44 \pm 1.23\%$ | $51.22 \pm 1.34\%$ | $42.88 \pm 1.35\%$ |
| | TADAM | $56.58 \pm 1.34\%$ | $33.34 \pm 1.27\%$ | $53.24 \pm 1.33\%$ | $43.06 \pm 1.33\%$ |
| | MAML | $53.94 \pm 1.45\%$ | $31.66 \pm 1.31\%$ | $51.37 \pm 1.38\%$ | $42.12 \pm 1.36\%$ |
| | MetaSGD | $55.58 \pm 1.43\%$ | $32.38 \pm 1.32\%$ | $52.99 \pm 1.36\%$ | $41.74 \pm 1.34\%$ |
| | BMAML | $54.89 \pm 1.48\%$ | $32.53 \pm 1.33\%$ | $53.63 \pm 1.37\%$ | $42.50 \pm 1.33\%$ |
| | MT-Net | $58.72 \pm 1.43\%$ | $32.80 \pm 1.35\%$ | $47.72 \pm 1.46\%$ | $43.11 \pm 1.42\%$ |
| | MUMOMAML | $56.82 \pm 1.49\%$ | $33.81 \pm 1.36\%$ | $53.14 \pm 1.39\%$ | $42.22 \pm 1.40\%$ |
| | HSML | $60.98 \pm 1.50\%$ | $35.01 \pm 1.36\%$ | $57.38 \pm 1.40\%$ | $44.02 \pm 1.39\%$ |
| | **ARML** | $\mathbf{62.33 \pm 1.47\%}$ | $\mathbf{35.65 \pm 1.40\%}$ | $\mathbf{58.56 \pm 1.41\%}$ | $\mathbf{44.82 \pm 1.38\%}$ |
| 5-way 5-shot | VERSA | $65.86 \pm 0.73\%$ | $37.46 \pm 0.65\%$ | $62.81 \pm 0.66\%$ | $48.03 \pm 0.78\%$ |
| | ProtoNet | $68.67 \pm 0.72\%$ | $45.21 \pm 0.67\%$ | $65.29 \pm 0.68\%$ | $51.27 \pm 0.81\%$ |
| | TapNet | $69.07 \pm 0.74\%$ | $45.54 \pm 0.68\%$ | $67.16 \pm 0.66\%$ | $51.08 \pm 0.80\%$ |
| | TADAM | $69.13 \pm 0.75\%$ | $45.78 \pm 0.65\%$ | $69.87 \pm 0.66\%$ | $53.15 \pm 0.82\%$ |
| | MAML | $68.52 \pm 0.79\%$ | $44.56 \pm 0.68\%$ | $66.18 \pm 0.71\%$ | $51.85 \pm 0.85\%$ |
| | MetaSGD | $67.87 \pm 0.74\%$ | $45.49 \pm 0.68\%$ | $66.84 \pm 0.70\%$ | $52.51 \pm 0.81\%$ |
| | BMAML | $69.01 \pm 0.74\%$ | $46.06 \pm 0.69\%$ | $65.74 \pm 0.67\%$ | $52.43 \pm 0.84\%$ |
| | MT-Net | $69.22 \pm 0.75\%$ | $46.57 \pm 0.70\%$ | $63.03 \pm 0.69\%$ | $53.49 \pm 0.83\%$ |
| | MUMOMAML | $70.49 \pm 0.76\%$ | $45.89 \pm 0.69\%$ | $67.31 \pm 0.68\%$ | $53.96 \pm 0.82\%$ |
| | HSML | $71.68 \pm 0.73\%$ | $48.08 \pm 0.69\%$ | $73.49 \pm 0.68\%$ | $56.32 \pm 0.80\%$ |
| | **ARML** | $\mathbf{73.34 \pm 0.70\%}$ | $\mathbf{49.67 \pm 0.67\%}$ | $\mathbf{74.88 \pm 0.64\%}$ | $\mathbf{57.55 \pm 0.82\%}$ |

different number of vertices in meta-knowledge graph in Appendix G.2 and conduct a comparison with HSML about the learned structure in Appendix G.3.

To analyze the learned meta-knowledge graph, for each subdataset, we randomly select one task as exemplary (see Figure 9 of Appendix G.4 for more cases). For each task, in the left part of Figure 3, we show the similarity heatmap between prototypes and vertices in meta-knowledge graph, where deeper color means higher similarity. V0-V8 and P1-P5 denote the different vertices and prototypes, respectively. The meta-knowledge graph is also illustrated in the right part. Similar to the graph in 2D regression, we set a threshold to filter links with low similarity and illustrate the rest of them. First, we can see that the V1 is mainly activated by bird and aircraft (including all filters), which may reflect the shape similarity between bird and aircraft. Second, V2, V3, V4 are firstly activated by texture and they form a loop in the meta-knowledge graph. Especially, V2 also benefits images with blur and pencil filters. Thus, V2 may represent the main texture and facilitate the training process on other subdatasets. The meta-knowledge graph also shows the importance of V2 since it is connected with almost all other vertices. Third, when we use blur filter, in most cases (bird blur, texture blur, fungi blur), V7 is activated. Thus, V7 may show the similarity of images with blur filter. In addition, the connection between V7 and V2 and V3 show that classifying blur images may depend on the texture information. Fourth, V6 (activated by aircraft mostly) connects with V2 and V3, justifying the importance of texture information to classify the aircrafts.

## 6 CONCLUSION

In this paper, to improve the effectiveness of meta-learning for handling heterogeneous task, we propose a new framework called ARML, which automatically extract relation across tasks and construct a meta-knowledge graph. When a new task comes in, it can quickly find the most relevant relations through the meta-knowledge graph and use this knowledge to facilitate its training process. The experiments demonstrate the effectiveness of our proposed algorithm.

In the future, we plan to investigate the problem in the following directions: (1) we are interested to investigate the more explainable semantic meaning in the meta-knowledge graph on this problem; (2)

Table 2: Overall few-shot classification results (accuracy $\pm$ 95% confidence) on Art-Multi dataset.

| Settings | Algorithms | Avg. Original | Avg. Blur | Avg. Pencil |
|---|---|---|---|---|
| 5-way, 1-shot | VERSA | $43.91 \pm 1.35\%$ | $41.98 \pm 1.35\%$ | $38.70 \pm 1.33\%$ |
| | Protonet | $42.08 \pm 1.34\%$ | $40.51 \pm 1.37\%$ | $36.24 \pm 1.35\%$ |
| | TapNet | $42.15 \pm 1.36\%$ | $41.16 \pm 1.34\%$ | $37.25 \pm 1.33\%$ |
| | TADAM | $44.73 \pm 1.33\%$ | $42.44 \pm 1.35\%$ | $39.02 \pm 1.34\%$ |
| | MAML | $42.70 \pm 1.35\%$ | $40.53 \pm 1.38\%$ | $36.71 \pm 1.37\%$ |
| | MetaSGD | $44.21 \pm 1.38\%$ | $42.36 \pm 1.39\%$ | $37.21 \pm 1.39\%$ |
| | MT-Net | $43.94 \pm 1.40\%$ | $41.64 \pm 1.37\%$ | $37.79 \pm 1.38\%$ |
| | BMAML | $43.66 \pm 1.36\%$ | $41.08 \pm 1.35\%$ | $37.28 \pm 1.39\%$ |
| | MUMOMAML | $45.63 \pm 1.39\%$ | $41.59 \pm 1.38\%$ | $39.24 \pm 1.36\%$ |
| | HSML | $45.68 \pm 1.37\%$ | $42.62 \pm 1.38\%$ | $39.78 \pm 1.36\%$ |
| | **ARML** | $\mathbf{47.92 \pm 1.34\%}$ | $\mathbf{44.43 \pm 1.34\%}$ | $\mathbf{41.44 \pm 1.34\%}$ |
| 5-way, 5-shot | VERSA | $55.03 \pm 0.71\%$ | $53.41 \pm 0.70\%$ | $47.93 \pm 0.70\%$ |
| | Protonet | $58.12 \pm 0.74\%$ | $55.07 \pm 0.73\%$ | $50.15 \pm 0.74\%$ |
| | TapNet | $57.77 \pm 0.73\%$ | $55.21 \pm 0.72\%$ | $50.68 \pm 0.71\%$ |
| | TADAM | $60.35 \pm 0.72\%$ | $58.36 \pm 0.73\%$ | $53.15 \pm 0.74\%$ |
| | MAML | $58.30 \pm 0.74\%$ | $55.71 \pm 0.74\%$ | $49.59 \pm 0.73\%$ |
| | MetaSGD | $57.82 \pm 0.72\%$ | $55.54 \pm 0.73\%$ | $50.24 \pm 0.72\%$ |
| | BMAML | $58.84 \pm 0.73\%$ | $56.21 \pm 0.71\%$ | $51.22 \pm 0.73\%$ |
| | MT-Net | $57.95 \pm 0.74\%$ | $54.65 \pm 0.73\%$ | $49.18 \pm 0.73\%$ |
| | MUMOMAML | $58.60 \pm 0.75\%$ | $56.29 \pm 0.72\%$ | $51.15 \pm 0.73\%$ |
| | HSML | $60.63 \pm 0.73\%$ | $57.91 \pm 0.72\%$ | $53.93 \pm 0.72\%$ |
| | **ARML** | $\mathbf{61.78 \pm 0.74\%}$ | $\mathbf{58.73 \pm 0.75\%}$ | $\mathbf{55.27 \pm 0.73\%}$ |

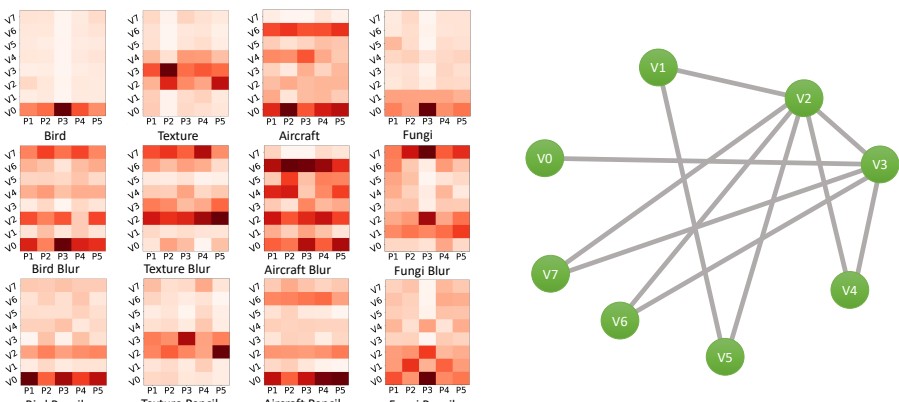

Figure 3: Interpretation of meta-knowledge graph on Art-Multi dataset. For each subdataset, we randomly select one task from them. In the left, we show the similarity heatmap between prototypes (P0-P5) and meta-knowledge vertices (V0-V7). In the right part, we show the meta-knowledge graph.

we plan to extend the ARML to the continual learning scenario where the structure of meta-knowledge graph will change over time; (3) our proposed model focuses on tasks where the feature space, the label space are shared. We plan to explore the relational structure on tasks with different feature and label spaces.

ACKNOWLEDGMENTS

The work was supported in part by NSF awards #1652525 and #1618448. The views and conclusions contained in this paper are those of the authors and should not be interpreted as representing any funding agencies.

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

## A    ALGORITHM IN META-TESTING PROCESS

---
**Algorithm 2** Meta-Testing Process of ARML
---
**Require:** Training data $\mathcal{D}_t^{tr}$ of a new task $\mathcal{T}_t$
 1: Construct the prototype-based relational graph $\mathcal{R}_t$ by computing prototype in equation 2 and weight in equation 4
 2: Compute the similarity between each prototype and meta-knowledge vertice in equation 6 and construct the super-graph $\mathcal{S}_t$
 3: Apply GNN on super-graph $\mathcal{S}_t$ and get the updated prototype representation $\hat{\mathbf{C}}_{\mathcal{R}_t}$
 4: Aggregate $\mathbf{C}_{\mathcal{R}_t}$ in equation 8, $\hat{\mathbf{C}}_{\mathcal{R}_t}$ in equation 9 and get the representations $\mathbf{q}_t, \mathbf{t}_t$
 5: Compute the task-specific initialization $\theta_{0t}$ in equation 10
 6: Update parameters $\theta_t = \theta_{0t} - \alpha \nabla_\theta \mathcal{L}(f_\theta, \mathcal{D}_t^{tr})$
---

## B  HYPERPARAMETERS SETTINGS

### B.1  2D REGRESSION

In 2D regression problem, we set the inner-loop stepsize (i.e., $\alpha$) and outer-loop stepsize (i.e., $\beta$) as 0.001 and 0.001, respectively. The embedding function $\mathcal{E}$ is set as one layer with 40 neurons. The autoencoder aggregator is constructed by the gated recurrent structures. We set the meta-batch size as 25 and the inner loop gradient steps as 5.

### B.2  FEW-SHOT IMAGE CLASSIFICATION

In few-shot image classification, for both Plain-Multi and Art-Multi datasets, we set the corresponding inner stepsize (i.e., $\alpha$) as 0.001 and the outer stepsize (i.e., $\beta$) as 0.01. For the embedding function $\mathcal{E}$, we employ two convolutional layers with $3 \times 3$ filters. The channel size of these two convolutional layers are 32. After convolutional layers, we use two fully connected layers with 384 and 128 neurons for each layer. Similar to the hyperparameter settings in 2D regression, the autoencoder aggregator is constructed by the gated recurrent structures, i.e., $AG^t$, $AG^t_{dec}$ $AG^q$, $AG^q_{dec}$ are all GRUs. The meta-batch size is set as 4. For the inner loop, we use 5 gradient steps.

### B.3  DETAILED BASELINE SETTINGS

For the gradient-based baselines (i.e., MAML, MetaSGD, MT-Net, BMAML. MUMOMAML, HSML), we use the same inner loop stepsize and outer loop stepsize rate as our ARML. As for non-parametric based meta-learning algorithms, both TADAM and Prototypical network, we use the same meta-training and meta-testing process as gradient-based models. Additionally, TADAM uses the same embedding function $\mathcal{E}$ as ARML for fair comparison (i.e., similar expressive ability).

## C  ADDITIONAL DISCUSSION OF DATASETS

In this dataset, we use pencil and blur filers to change the task distribution. To investigate the effect of pencil and blur filters, we provide one example in Figure 4. We can observe that different filters result in different data distributions. All used filters are provided by OpenCV[1].

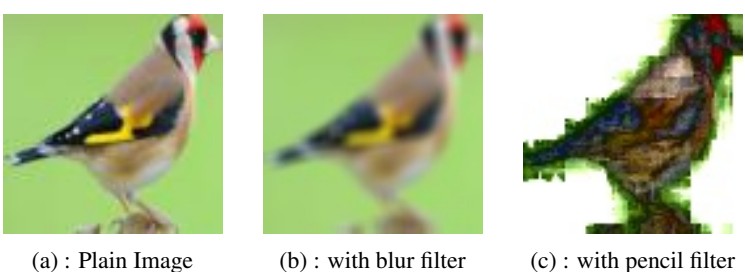

(a) : Plain Image          (b) : with blur filter          (c) : with pencil filter

Figure 4: Effect of different filters.

## D  RESULTS ON MINIIMAGENET AND TIEREDIMAGENET

For miniimagenet and tieredImagenet, since it do not have the characteristic of task heterogeneity, we show the results in Table 3 and Table 4, respectively. In this table, we compare our model with other gradient-based meta-learning models (the top baselines are globally shared models and the bottom baselines are task-specific models). Similar to (Finn et al., 2018), we also apply the standard 4-block convolutional layers for each baseline. For MT-Net on MiniImagenet, we use the reported results in (Yao et al., 2019b), which control the model with the same expressive power. Most task-specific models including ARML achieve the similar performance on the standard benchmark due to the homogeneity between tasks.

---

[1]https://opencv.org/

Table 3: Performance comparison on the 5-way, 1-shot MiniImagenet dataset.

| Algorithms | 5-way 1-shot Accuracy |
|---|---|
| MAML (Finn et al., 2017) | $48.70 \pm 1.84\%$ |
| LLAMA (Finn & Levine, 2018) | $49.40 \pm 1.83\%$ |
| Reptile (Nichol & Schulman, 2018) | $49.97 \pm 0.32\%$ |
| MetaSGD (Li et al., 2017) | $50.47 \pm 1.87\%$ |
| MT-Net (Lee & Choi, 2018) | $49.75 \pm 1.83\%$ |
| MUMOMAML (Vuorio et al., 2018) | $49.86 \pm 1.85\%$ |
| HSML (Yao et al., 2019b) | $50.38 \pm 1.85\%$ |
| PLATIPUS (Finn et al., 2018) | $50.13 \pm 1.86\%$ |
| ARML | $50.42 \pm 1.73\%$ |

Table 4: Performance comparison on the 5-way, 1-shot tieredImagenet dataset.

| Algorithms | 5-way 1-shot Accuracy |
|---|---|
| MAML (Finn et al., 2017) | $51.37 \pm 1.80\%$ |
| Reptile (Nichol & Schulman, 2018) | $49.41 \pm 1.82\%$ |
| MetaSGD (Li et al., 2017) | $51.48 \pm 1.79\%$ |
| MT-Net (Lee & Choi, 2018) | $51.95 \pm 1.83\%$ |
| MUMOMAML (Vuorio et al., 2018) | $52.59 \pm 1.80\%$ |
| HSML (Yao et al., 2019b) | $52.67 \pm 1.85\%$ |
| ARML | $52.91 \pm 1.83\%$ |

## E  ADDITIONAL RESULTS OF FEW-SHOT IMAGE CLASSIFICATION

We provide the full results table of Art-Multi Dataset in Table 8. In this table, we can see our proposed ARML outperforms almost all baselines in every sub-datasets.

## F  ABLATION STUDY

In this section, we perform the ablation study of the proposed ARML to demonstrate the effectiveness of each component. The results of ablation study on 5-way, 5-shot scenario for Art-Multi and Plain-Multi datasets are presented in Table 5 and Table 6, respectively. Specifically, to show the effectiveness of prototype-based relational graph, in ablation I, we apply the mean pooling to aggregate each sample and then feed it to interact with meta-knowledge graph. In ablation II, we use all samples to construct the sample-level relational graph without constructing prototype. In ablation III, we remove the links between prototypes. Compared with ablation I, II and III, the better performance of ARML shows that structuring samples can (1) better handling the underlying relations (2) alleviating the effect of potential anomalies by structuring samples as prototypes.

In ablation IV, we remove the meta-knowledge graph and use the prototype-based relational graph with aggregator $\mathrm{AG}^q$ as the task representation. The better performance of ARML demonstrates the effectiveness of meta-knowledge graph for capturing the relational structure and facilitating the classification performance. We further remove the reconstruction loss in ablation V and replace the encoder/decoder structure as MLP in ablation VI. The results demonstrate that the autoencoder structure benefits the process of task representation learning and selected encoder and decoder.

In ablation VII, we share the gate value within each filter in Convolutional layers. Compared with VII, the better performance of ARML indicates the benefit of customized gate for each parameter. In ablation VIII and IX, we change the modulate function to Film (Perez et al., 2018) and tanh, respectively. We can see that ARML is not very sensitive to the modulating activation, and Sigmoid function is slightly better in most cases.

Table 5: Full evaluation results of model ablation study on Art-Multi dataset. B, T, A, F represent bird, texture, aircraft, fungi, respectively. Plain means original image.

| Model | B Plain | B Blur | B Pencil | T Plain | T Blur | T Pencil |
|---|---|---|---|---|---|---|
| I. no prototype-based graph | 72.08% | 71.06% | 66.83% | 45.23% | 39.97% | 41.67% |
| II. no prototype | 72.99% | 70.92% | 67.19% | 45.17% | 40.05% | 41.04% |
| III. no prototype links | 72.53% | 71.09% | 67.11% | 45.08% | 40.12% | 41.01% |
| IV. no meta-knowledge graph | 70.79% | 69.53% | 64.87% | 43.37% | 39.86% | 41.23% |
| V. no reconstruction loss | 70.82% | 69.87% | 65.32% | 44.02% | 40.18% | 40.52% |
| VI. replace encoder/decoder as MLP | 71.36% | 70.25% | 66.38% | 44.18% | 39.97% | 41.85% |
| VII. share the gate within Conv. filter | 72.03% | 68.15% | 65.23% | 43.98% | 40.13% | 39.71% |
| VIII. tanh | 72.70% | 69.53% | 66.85% | **45.81**% | **40.79**% | 38.64% |
| IX. film | 71.52% | 68.70% | 64.23% | 43.83% | 40.52% | 39.49% |
| ARML | **73.05**% | **71.31**% | **67.14**% | 45.32% | 40.15% | **41.98**% |
| Model | A Plain | A Blur | A Pencil | F Plain | F Blur | F Pencil |
| I. no prototype-based graph | 70.06% | 68.02% | 60.66% | 55.81% | 54.39% | 50.01% |
| II. no prototype | 71.10% | 67.59% | 61.07% | 56.11% | 54.82% | 49.95% |
| III. no prototype links | 71.56% | 67.91% | 60.83% | 55.76% | 54.60% | 50.08% |
| IV. no meta-knowledge graph | 69.97% | 68.03% | 59.72% | 55.84% | 53.72% | 48.91% |
| V. no reconstruction loss | 66.83% | 65.73% | 55.98% | 54.62% | 53.02% | 48.01% |
| VI. replace encoder/decoder as MLP | 70.93% | 68.12% | 60.95% | 56.02% | 53.83% | 50.22% |
| VII. share the gate within Conv. filter | 71.25% | 67.49% | 58.09% | 55.36% | 54.25% | 49.90% |
| VIII. tanh | **73.96**% | **69.70**% | 60.75% | **56.87**% | 54.30% | 49.82% |
| IX. film | 69.13% | 66.93% | 55.59% | 55.77% | 53.72% | 48.92% |
| ARML | 71.89% | 68.59% | **61.41**% | 56.83% | **54.87**% | **50.53**% |

Table 6: Results of Model Ablation (5-way, 5-shot results) on Plain-Multi dataset.

| Ablation Models | Bird | Texture | Aircraft | Fungi |
|---|---|---|---|---|
| I. no prototype-based graph | $71.96 \pm 0.72\%$ | $48.79 \pm 0.67\%$ | $74.02 \pm 0.65\%$ | $56.83 \pm 0.80\%$ |
| II. no prototype | $72.86 \pm 0.74\%$ | $49.03 \pm 0.69\%$ | $74.36 \pm 0.65\%$ | $57.02 \pm 0.81\%$ |
| III. no prototype links | $72.53 \pm 0.72\%$ | $49.25 \pm 0.68\%$ | $74.46 \pm 0.64\%$ | $57.10 \pm 0.81\%$ |
| IV. no meta-knowledge graph | $71.23 \pm 0.75\%$ | $47.96 \pm 0.68\%$ | $73.71 \pm 0.69\%$ | $55.97 \pm 0.82\%$ |
| V. no reconstruction loss | $70.99 \pm 0.74\%$ | $48.03 \pm 0.69\%$ | $69.86 \pm 0.66\%$ | $55.78 \pm 0.83\%$ |
| VI. replace encoder/decoder as MLP | $72.36 \pm 0.72\%$ | $48.93 \pm 0.67\%$ | $74.28 \pm 0.65\%$ | $56.91 \pm 0.83\%$ |
| VII. share the gate within Conv. filter | $72.83 \pm 0.72\%$ | $48.66 \pm 0.68\%$ | $74.13 \pm 0.66\%$ | $56.83 \pm 0.81\%$ |
| VIII. tanh | $\mathbf{73.45 \pm 0.71}\%$ | $49.23 \pm 0.66\%$ | $74.39 \pm 0.65\%$ | $57.38 \pm 0.80\%$ |
| IX. film | $72.95 \pm 0.73\%$ | $49.18 \pm 0.69\%$ | $73.82 \pm 0.68\%$ | $56.89 \pm 0.80\%$ |
| ARML | $73.34 \pm 0.70\%$ | $\mathbf{49.67 \pm 0.67}\%$ | $\mathbf{74.88 \pm 0.64}\%$ | $\mathbf{57.55 \pm 0.82}\%$ |

# G  ADDITIONAL ANALYSIS OF META-KNOWLEDGE GRAPH

## G.1  QUALITATIVE ANALYSIS ON PLAIN-MULTI DATASET

Then, we analyze the meta-knowledge graph on Plain-Multi dataset by visualizing the learned meta-knowledge graph on Plain-Multi dataset (as shown in Figure 5). In this figure, we can see that different subdatasets activate different vertices. Specifically, V2, which is mainly activated by texture, plays a significantly important role in aircraft and fungi. Thus, V2 connects with V3 and V1 in the meta-knowledge graph, which is mainly activated by fungi and aircraft, respectively. In addition, V0 is also activated by aircraft because of the similar contour between aircraft and bird. Furthermore, in meta-knowledge graph, V0 connects with V3, which shows the similarity of environment between bird images and fungi images.

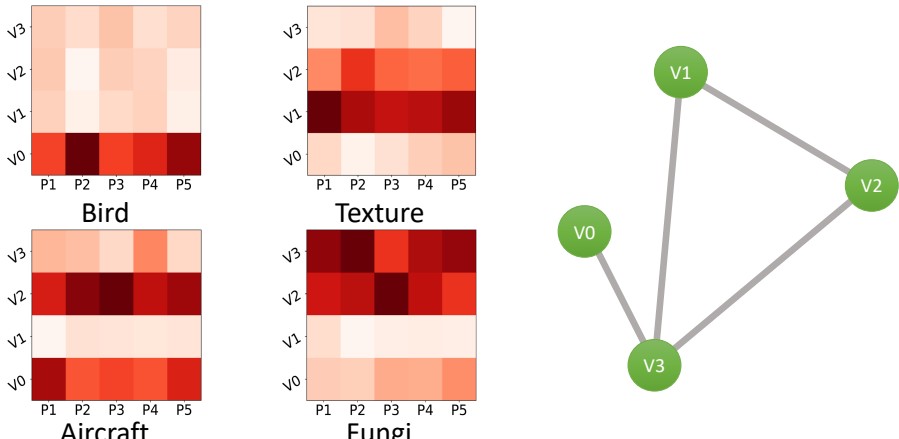

Figure 5: Interpretation of meta-knowledge graph on Plain-Multi dataset. For each subdataset, one task is randomly selected from them. In the left figure, we show the similarity heatmap between prototypes (P1-P5) and meta-knowledge vertices (denoted as E1-E4), where deeper color means higher similarity. In the right part, we show the meta-knowledge graph, where a threshold is also set to filter low similarity links.

## G.2 PERFORMANCE V.S. VERTICE NUMBERS

We first investigate the impact of vertice numbers in meta-knowledge graph. The results of Art-Multi (5-way, 5-shot) are shown in Table 7. From the results, we can notice that the performance saturates as the number of vertices around 8. One potential reason is that 8 vertices are enough to capture the potential relations. If we have a larger datasets with more complex relations, more vertices may be needed. In addition, if the meta-knowledge graph does not have enough vertices, the worse performance suggests that the graph may not capture enough relations across tasks.

Table 7: Full evaluation results of performance v.s. # vertices of meta-knowledge graph on Art-Multi. B, T, A, F represent bird, texture, aircraft, fungi, respectively. Plain means original image.

| # of Vertices | B Plain | B Blur | B Pencil | T Plain | T Blur | T Pencil |
|---|---|---|---|---|---|---|
| 4 | 72.29% | 70.36% | 67.88% | 45.37% | 41.05% | 41.43% |
| 8 | 73.05% | 71.31% | 67.14% | 45.32% | 40.15% | 41.98% |
| 12 | 73.45% | 70.64% | 67.41% | 44.53% | 41.41% | 41.05% |
| 16 | 72.68% | 70.18% | 68.34% | 45.63% | 41.43% | 42.18% |
| 20 | 73.41% | 71.07% | 68.64% | 46.26% | 41.80% | 41.61% |

| # of Vertices | A Plain | A Blur | A Pencil | F Plain | F Blur | F Pencil |
|---|---|---|---|---|---|---|
| 4 | 70.98% | 67.36% | 60.46% | 56.07% | 53.77% | 50.08% |
| 8 | 71.89% | 68.59% | 61.41% | 56.83% | 54.87% | 50.53% |
| 12 | 71.78% | 67.26% | 60.97% | 56.87% | 55.14% | 50.86% |
| 16 | 71.96% | 68.55% | 61.14% | 56.76% | 54.54% | 49.41% |
| 20 | 72.02% | 68.29% | 60.59% | 55.95% | 54.53% | 50.13% |

## G.3 DISCUSSION BETWEEN ARML AND HSML

In this part, we provide the case study to visualize the task structure of HSML and ARML. HSML is one of representative task-specific meta-learning methods, which adapts transferable knowledge by introducing a task-specific representation. It proposes a tree structure to learn the relations between tasks. However, the structure requires massive labor efforts to explore the optimal structure. By contrast, ARML automatically learn the relation across tasks by introducing the knowledge graph. In addition, ARML fully exploring there types of relations simultaneously, i.e., the prototype-prototype, prototype-knowledge and knowledge-knowledge relations.

To compare these two models, we show the case studies of HSML and ARML in Figure 6 and Figure 7. For tasks sampled from bird, bird blur, aircraft and aircraft blur are selected for this comparison. Following case study settings in the original paper (Yao et al., 2019b), for each task, we show the soft-assignment probability to each cluster and the learned hierarchical structure. For ARML, like 3, we show the learned meta-knowledge and the similarity heatmap between prototypes and meta-knowledge vertices. In this figures we can observe that ARML constructs relations in a more flexible way by introducing the graph structure. More specifically, while HSML activate relevant node in a fixed two-layer hierarchical way, ARML provides more possibilities to leverage previous learned tasks by leveraging prototypes and the learned meta-knowledge graph.

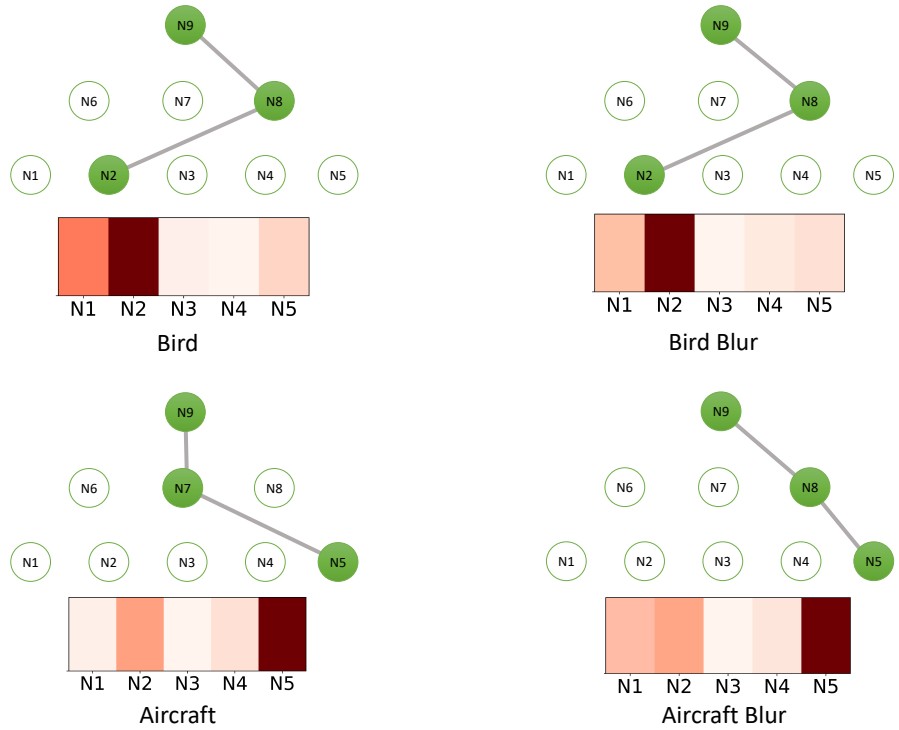

Figure 6: Case study for learned structure of HSML. For each task, the top heatmap shows the soft-assignment probability with each cluster and the top tree show the learned hierarchical structure.

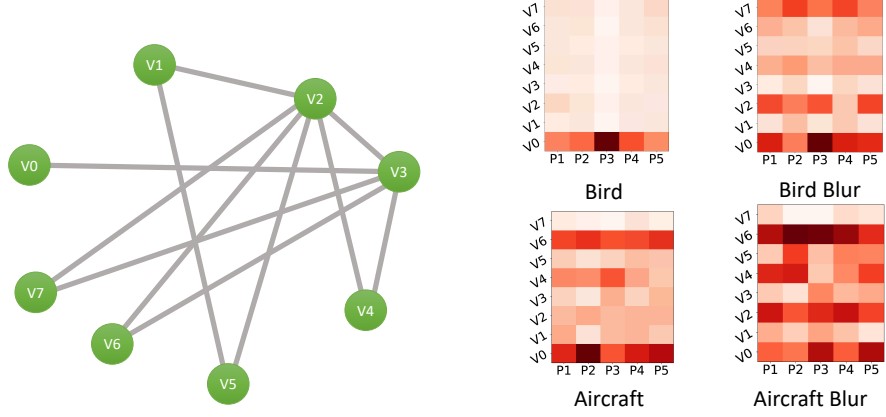

Figure 7: Case study for learned structure of ARML. Both the learned graph and the similarity heatmap between prototypes and meta-knowledge vertices are illustrated.

## G.4 ADDITIONAL CASES OF META-KNOWLEDGE GRAPH ANALYSIS

We provide additional case study in this section. In Figure 8, we show the cases of 2D regression and the additional cases of Art-Multi are illustrated in Figure 9. We can see the additional cases also support our observations and interpretations.

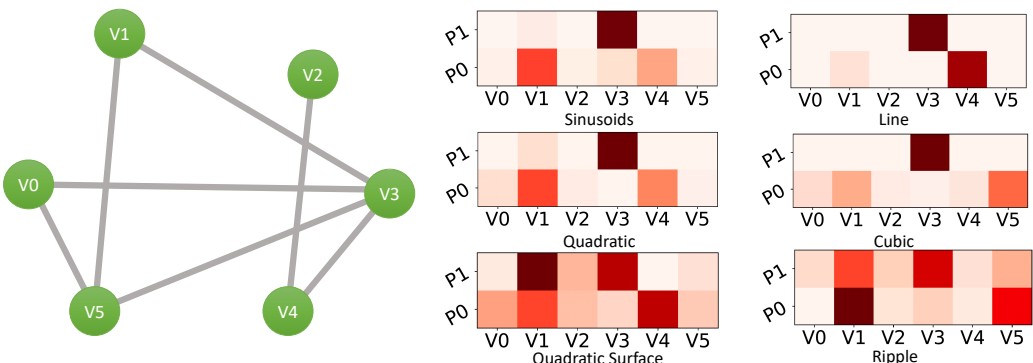

Figure 8: Additional cases on 2D regression.

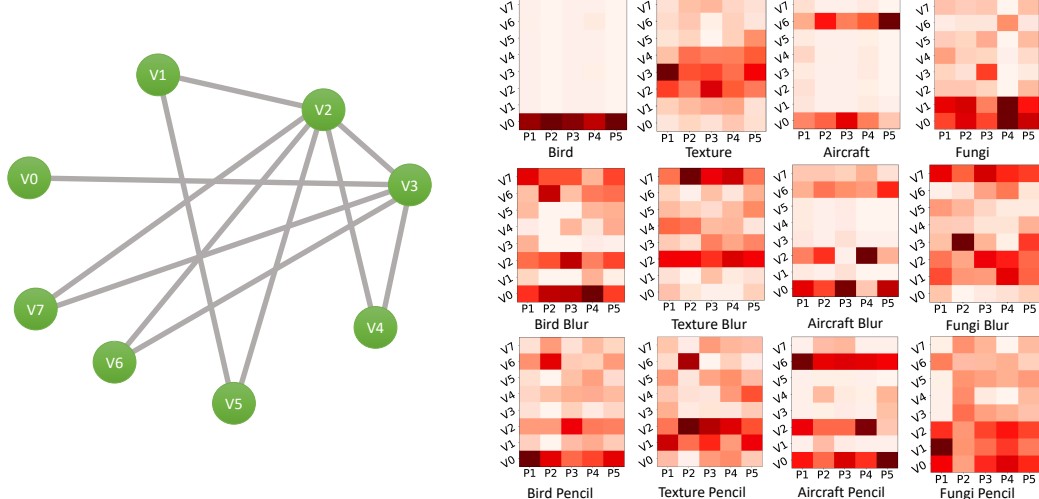

Figure 9: Additional cases of Art-Multi.

Table 8: Full results on Art-Multi dataset. In this table, B, T, A, F represent bird, texture, aircraft, fungi, respectively. Plain means original image.

| Settings | Algorithms | B Plain | B Blur | B Pencil | T Plain | T Blur | T Pencil | A Plain | A Blur | A Pencil | F Plain | F Blur | F Pencil |
|---|---|---|---|---|---|---|---|---|---|---|---|---|---|
| 5-way 1-shot | VERSA | 52.46% | 51.65% | 48.80% | 30.03% | 29.10% | 28.74% | 51.03% | 47.89% | 41.07% | 42.13% | 39.26% | 36.20% |
| | ProtoNet | 53.67% | 50.98% | 46.66% | 31.37% | 29.08% | 28.48% | 45.54% | 43.94% | 35.49% | 37.71% | 38.00% | 34.36% |
| | TapNet | 53.30% | 51.14% | 47.76% | 31.56% | 29.48% | 29.08% | 46.18% | 44.50% | 36.66% | 37.54% | 39.50% | 35.46% |
| | TADAM | 54.76% | 52.18% | 48.85% | 32.03% | 29.90% | 30.82% | 50.42% | 47.59% | 40.17% | 41.73% | 40.09% | 36.27% |
| | MAML | 55.27% | 52.62% | 48.58% | 30.57% | 28.65% | 28.39% | 45.59% | 42.24% | 34.52% | 39.37% | 38.58% | 35.38% |
| | MetaSGD | 55.23% | 53.08% | 48.18% | 29.28% | 28.70% | 28.38% | 51.24% | 47.29% | 35.98% | 41.08% | 40.38% | 36.30% |
| | BMAML | 56.71% | 52.87% | 47.83% | 31.02% | 29.11% | 29.69% | 46.83% | 42.68% | 36.08% | 40.09% | 39.66% | 35.51% |
| | MT-Net | 56.99% | 54.21% | 50.25% | 32.13% | 29.63% | 29.23% | 43.64% | 40.08% | 33.73% | 43.02% | 42.64% | 37.96% |
| | MUMOMAML | 57.73% | 53.18% | 50.96% | 31.88% | 29.72% | 29.90% | 49.95% | 43.36% | 39.61% | 42.97% | 40.08% | 36.52% |
| | HSML | 58.15% | 53.20% | 51.09% | 32.01% | 30.21% | 30.17% | 49.98% | 45.79% | 40.87% | 42.58% | 41.29% | 37.01% |
| | **ARML** | **59.67%** | **54.89%** | **52.97%** | **32.31%** | **30.77%** | **31.51%** | **51.99%** | **47.92%** | **41.93%** | **44.69%** | **42.13%** | **38.36%** |
| 5-way 5-shot | VERSA | 66.28% | 65.12% | 60.76% | 38.85% | 35.49% | 33.83% | 64.82% | 62.73% | 53.60% | 51.18% | 50.30% | 43.54% |
| | ProtoNet | 70.42% | 67.90% | 61.82% | 44.78% | 38.43% | 38.40% | 65.84% | 63.41% | 54.08% | 51.45% | 50.56% | 46.33% |
| | TapNet | 68.60% | 68.03% | 62.69% | 43.41% | 37.86% | 38.60% | 65.16% | 64.29% | 54.73% | 53.92% | 50.66% | 46.69% |
| | TADAM | 70.08% | 69.05% | 65.45% | 44.93% | 41.80% | 40.18% | 70.35% | 68.56% | 59.09% | 56.04% | 54.04% | 47.85% |
| | MAML | 71.51% | 68.65% | 63.93% | 42.96% | 39.59% | 38.87% | 64.68% | 62.54% | 49.20% | 54.08% | 52.02% | 46.39% |
| | MetaSGD | 71.31% | 68.73% | 64.33% | 41.89% | 37.79% | 37.91% | 64.88% | 63.36% | 52.31% | 53.18% | 52.26% | 46.43% |
| | BMAML | 71.66% | 68.51% | 64.99% | 43.18% | 39.83% | 39.76% | 66.57% | 63.33% | 51.91% | 53.96% | 53.18% | 48.21% |
| | MT-Net | 71.18% | 69.29% | 68.28% | 43.23% | 39.42% | 39.20% | 63.39% | 58.29% | 46.12% | 54.01% | 51.70% | 47.02% |
| | MUMOMAML | 71.57% | 70.50% | 64.57% | 44.57% | 40.31% | 40.07% | 63.36% | 61.55% | 52.17% | 54.89% | 52.82% | 47.79% |
| | HSML | 71.75% | 69.31% | 65.62% | 44.68% | 40.13% | 41.33% | 70.12% | 67.63% | 59.40% | 55.97% | 54.60% | 49.40% |
| | **ARML** | **73.05%** | **71.31%** | **67.14%** | **45.32%** | 40.15% | **41.98%** | **71.89%** | **68.59%** | **61.41%** | **56.83%** | **54.87%** | **50.53%** |

