# OpenReview forum: "Automated Relational Meta-learning"
_ICLR.cc/2020/Conference — Accept (Poster)_

### Official Review · AnonReviewer2 · 2019-10-22
**Official Blind Review #2**

**Rating:** 8

**Review:**

This paper proposed a knowledge-based meta-learning framework, called ARML(Automated Relational Metal-Learning) that automatically extracts cross-task relations and constructs a meta-knowledge graph. ARML wanted to solve the task heterogeneity problem in meta-learning through knowledge graph learning using graph neural networks. To do this, the authors introduced a framework consisting of (1) finding a prototype-based relational structure, (2) constructing a meta-knowledge graph, and (3) adapting the task-specific knowledge. Experimental results show that the proposed algorithm outperforms other competitive algorithms in few-shot learning tasks, which is justified by experimentally showing that the learned meta-knowledge graph has a meaningful interpretation.

The paper was well-motivated and well-written, which made it very interesting to read. Looking at the task heterogeneity problem of meta-learning as a knowledge graph learning problem is the most important contribution of this paper. Since then, the framework's proposal to learn it as a graph neural network is a very natural extension, which can greatly increase the performance of existing few-shot learning tasks.

The question here is whether the meta-learning method for finding relational structures through knowledge graphs is the first one proposed in this paper. The paper "Few-shot learning with graph neural networks, ICLR-2018" performed the few-shot learning task with very similar motivation. What is the difference compared to this paper?

And as mentioned in the paper, HSML is the closest study to ARML in that it considers high-level relations between cross-tasks. The reviewer is very curious about the qualitative comparison of the high-level structures found by the two algorithms, and I confident that this comparison will enrich the paper.

**Experience Assessment:**

I have published one or two papers in this area.

**Review Assessment: Checking Correctness Of Derivations And Theory:**

I assessed the sensibility of the derivations and theory.

**Review Assessment: Checking Correctness Of Experiments:**

I assessed the sensibility of the experiments.

**Review Assessment: Thoroughness In Paper Reading:**

I read the paper at least twice and used my best judgement in assessing the paper.

---

> ### Author Response · Authors · 2019-11-12
> **Response to Review #2**
>
> Thanks a lot for the constructive comments and pointing out the potential confusion.
>
> Q1: Difference between ICLR 2018
> A1: Thank you for offering us a chance to explain the differences between the paper in ICLR’2018 and our approach. The major differences are listed as follows:
>     - ICLR 2018: Like other globally shared meta-learning models [Finn ICML’17;Snell NeurIPS’17], the goal of that paper is to learn a globally-shared meta-learner to facilitate the learning process on new tasks. They regard the few-shot learning problem as a semi-supervised learning task and adopt graph-based semi-supervised learning method to solve it. The contribution of this paper is to infer labels in test set by passing messages in a constructed graph. Specifically, the graph is constructed by treating each sample as a vertex. The edge weights are gauged by the embedding similarity between corresponding vertices. Then, they propagate the labels in the training set to infer unknown labels in the test set.
>     - Ours: We utilize a meta-knowledge graph in our model in order to enable relevant information retrieval from historical knowledge. In other words, instead of utilizing a graph to propagate label information in ICLR 2018, we aim to learn from the knowledge propagated from relevant summarized historical tasks to enhance the representation learning of the current task. To the best of our knowledge, we are the first to capture cross-task relationship by learning and leveraging meta-knowledge graph.
>
>
> Q2: Comparison of learned structure between HSML and ARML(ours)
> A2: The relation between tasks in HSML is expressed by a predefined tree structure. Thus, the setting of input structure requires external human prior knowledge, which involves massive labor efforts to explore the optimal structure. For instance, it requires careful setting with # layers and # nodes in each layer. On the contrary, ARML provides a fully automatic solution by introducing a graph to capture the task dependencies. Note that, the hierarchical structure is a special case in the graph.
>
> Furthermore, the tree structure in HSML requires the aggregation across prototype representations to be ready before querying the historical knowledge. Different from HSML, our graph structure is constructed with the goal of tapping into the input task with the historical knowledge by fully exploring the prototype-prototype, prototype-knowledge and knowledge-knowledge relations simultaneously. More specifically, the prototype representation is enriched by leveraging relevant information from the knowledge-knowledge structure. The task representation, which summarizes the internal categories and their pairwise relations, is more complete and comprehensive, as the information aggregation is conducted at the very last step with almost no information loss.
>
> In the revised version, we add the comparison of case studies between HSML and ARML in Appendix H.1. Four tasks sampled from bird, bird blur, aircraft, aircraft blur are selected for case studies (these four tasks are also used in the original analysis in Figure 3). For HSML, we follow the setting of case study in the original paper. In each task, we show the soft-assignment probability to each cluster and the activated clusters in the tree structure. For ARML, we show the learned structure and the similarity heatmap between prototypes and meta-knowledge vertices. From the visualized structures in different tasks, we can observe that our proposed model involves historical knowledge in a more flexible way. More specifically, while HSML activates the relevant clusters in a hierarchical way, ARML provides more possibilities to leverage summarized historical knowledge. Note that, a hierarchical/tree structure is a special case in a graph structure.
>
>
> References:
> [Finn ICML’17] Finn, Chelsea, Pieter Abbeel, and Sergey Levine. "Model-agnostic meta-learning for fast adaptation of deep networks." Proceedings of the 34th International Conference on Machine Learning-Volume 70. JMLR. org, 2017.
> [Snell NeurIPS’17] Snell, Jake, Kevin Swersky, and Richard Zemel. "Prototypical networks for few-shot learning." Advances in Neural Information Processing Systems. 2017.

---

### Official Review · AnonReviewer3 · 2019-10-23
**Official Blind Review #3**

**Rating:** 8

**Review:**

################################################################################
Summary:

The paper provides a interesting direction in the meta-learning filed. In particular, it proposes to enhance meta learning performance by fully exploring relations across multiple tasks. To capture such information, the authors develop a heterogeneity-aware meta-learning framework by introducing a novel architecture--meta-knowledge graph, which can dynamically find the most relevant structure for new tasks.


################################################################################
Reasons for score:

Overall, I vote for accepting. I like the idea of mining the relation between tasks and handle it by the proposed meta-knowledge graph. My major concern is about the clarity of the paper and some additional ablation models (see cons below). Hopefully the authors can address my concern in the rebuttal period.

################################################################################
Pros:

1. The paper takes one of the most important issue of meta-learning: task heterogeneity. For me, the problem itself is real and practical.

2. The proposed meta-knowledge graph is novel for capturing the relation between tasks and address the problem of task heterogeneity.
Graph structure provides a more flexible way of modeling relations.
The design for using the prototype-based relational graph to query the meta-knowledge graph is reasonable and interesting.

3. This paper provides comprehensive experiments, including both qualitative analysis and quantitative results,  to show the effectiveness of the proposed framework. The newly constructed Art-Multi dataset further enhances the difficulty of tasks and makes the performance more convincing.

################################################################################
Cons:

1. Although the proposed method provides several ablation studies, I still suggest the authors to conduct the following ablation studies to enhance the quality of the paper:
(1) It might be valuable to investigate the modulation function. In the paper, the authors compare sigmoid, tanh, and Film layer. Can the authors analyze the results by reducing the number of gating parameters in Eq. 10 by sharing the gate value of each filter in Conv layers?

(2) What is the performance of the proposed model by changing the type of aggregators?

2. For the autoencoder aggregator, it would be better to provide more details about it, which seems not very clear to me.

3. In the qualitative analysis (i.e., Figure 2 and Figure 3), the authors provide one visualization for each task. It would be more convincing if the authors can provide more cases in the rebuttal period.


################################################################################
Questions during rebuttal period:

Please address and clarify the cons above

################################################################################
Some typos:
(1) Table 7: I. no sample-level graph -> I. no prototype-based graph
(2) 5.1 Hyperparameter Settings: we try both sigmoid, tanh Film -> we try both sigmoid, tanh,
Film.
(3) parameteric -> parametric
(4) Table 2: Origninal -> original
(5) Section 4 first paragraph:  The enhanced prototype representation -> The enhanced prototype representations


Updates: Thanks for the authors' response. The newly added experimental results address my concerns. I believe this paper will provide new insights for this field and I recommend this paper to be accepted.


**Experience Assessment:**

I have published one or two papers in this area.

**Review Assessment: Checking Correctness Of Derivations And Theory:**

N/A

**Review Assessment: Checking Correctness Of Experiments:**

I carefully checked the experiments.

**Review Assessment: Thoroughness In Paper Reading:**

I read the paper thoroughly.

---

> ### Author Response · Authors · 2019-11-12
> **Response to Review #3**
>
> We greatly appreciate your comments and thoughtful suggestions. You may find our corresponding explanations and solutions below for the issues.
>
> Q1: More ablation studies
> A1: We’ve conducted the ablation studies and show the results of 5-way 5-shot as follows (see the revised paper for 5-way 1-shot results) as follows:
>     - We change the encode and decode model from GRU to MLP (ablation model VI in the revised paper):
>         - Results of Plain-Multi: bird 72.36±0.72% | texture 48.93±0.67% | aircraft 74.28±0.65% | fungi 56.91±0.83%
>         - Results of Art-Multi: ave. original 60.62±0.73% | ave. blur 58.04±0.72% | ave. pencil 54.85±0.72%
> The better performance of ARML indicates GRU may be a better choice due to its higher expressive power.
>     - The results of sharing gate within each filter in Conv layers are (ablation model VII in the revised paper):
>         - Results of Plain-Multi: bird 72.83±0.72% | texture 48.66±0.68% | aircraft 74.13±0.66% | fungi 56.83±0.81%
>         - Results of Art-Multi: ave. original 60.65±0.74% | ave. blur 57.51±0.75% | ave. pencil 53.23±0.74%
> Compared with ablation VII, ARML achieves better performance, indicating the benefits of the customized gate for each parameter.
>
>
> Q2: Detailed description of autoencoder aggregator
> A2: The autoencoder aggregator consists of one encoder and one decoder model. In this paper, we adopt GRU as the encoder model and decoder model. The input of GRU is [#prototype, #embeded dim]. Then, a mean pooling layer is applied to the output of GRU to calculate task representation [1, #embeded dim]. To enhance the learning stability, like [Srivastava ICML’15], the decoder is used to reversely reconstruct the input. The output of the decoder is still [#prototype, #embedd dim]. Then, the reconstruction error is calculated by the mean square loss between the output of the decoder and the input.
>
>
> Q3: More qualitative cases
> A3: We’ve added more qualitative cases in Appendix H.2. The observations are similar to the discussion in the paper (i.e., Figure 2 and 3). The results further support the motivation for constructing the meta-knowledge graph.
>
>
> References:
> [Srivastava ICML’15] Srivastava, Nitish, Elman Mansimov, and Ruslan Salakhudinov. "Unsupervised learning of video representations using lstms." International conference on machine learning. 2015.

---

### Official Review · AnonReviewer1 · 2019-10-23
**Official Blind Review #1**

**Rating:** 3

**Review:**

This paper mainly tackles the problem of heterogeneous tasks in meta-learning by proposing a new meta-learning framework ARML, which contains a module extracting relations across classes and a module representing meta-knowledge. When processing a new task, a graphical task representation is firstly constructed based on class prototypes, and then information propagation is conducted on a super-graph to find the most relevant meta-knowledge in the meta-knowledge graph. Ideally, the higher similarity between a prototype and a meta-knowledge node means the higher the correlation between a class and a specific type of meta-knowledge. In order to construct task-specific meta-learners, the authors utilize two auto-encoders to encode task representations with and without meta-knowledge graph. After that, a modulating function is applied to a set of shared parameters, which finishes the calculation of task-specific parameters. The authors empirically evaluated the proposed method on several datasets and it seems that ARML outperforms some compared methods.

This paper should be rejected. Firstly, the proposed method is not well motivated. It’s true that tasks in meta-learning may be sampled from a complex (or multi-modal) task distribution, but why to represent a task as a graph? I think the relation between tasks can be simply obtained from instances (CNN embeddings). Secondly, it’s hard to say the meta-knowledge graph can really capture knowledge with ‘exact meanings’ even though in some situations, a subset of nodes is activated and others are not.

Main arguments
1.	The whole framework is too complex and it’s hard to say every module in the framework really works even ablation study is done.
2.	The meta-knowledge graph lacks interpretability. From my perspective, it’s just a set of learnable parameters without any exact meanings. Authors tried to analyze the constructed meta-knowledge graph by some experiments, but these discussions are farfetched.

Things to improve the paper
1.	Simplify the proposed method.
2.	Make it clear why should we represent a task as a graph.
3.	Some most widely used benchmark datasets such as mini-imagenet and tiered-imagenet are not used. For a fair and convincing comparison, I suggest the authors test the proposed method on these benchmark datasets. Moreover, more methods should be compared.

**Experience Assessment:**

I have published one or two papers in this area.

**Review Assessment: Checking Correctness Of Derivations And Theory:**

N/A

**Review Assessment: Checking Correctness Of Experiments:**

I assessed the sensibility of the experiments.

**Review Assessment: Thoroughness In Paper Reading:**

I read the paper at least twice and used my best judgement in assessing the paper.

---

> ### Author Response · Authors · 2019-11-12
> **Response to Review #1, part 1/2**
>
> Thank you for your constructive and valuable comments. We’ve revised our paper following the suggestions and will explain your concerns in the following.
>
> Q1: Framework is too complex (“Things to improve the paper 1/Main argument 1”)
> A1: Each component of our model speaks for itself and they collectively fulfill the entire goal. We clarify the motivation of each major component and its corresponding ablation models:
>     - Prototype-based relational graph is used to summarize samples (use prototype) and extract the inner relation between prototypes. Ablation models I, II, and III demonstrate its contribution.
>     - Meta-knowledge graph is used to organize previous learned knowledge by extracting the relationship between previous tasks. Ablation models IV verify its effectiveness.
>     - Task-specific adaptation is used to customize the globally shared initialization by using task-specific information. The superior performance over globally shared models proves its ability to leverage task relation. Ablation models VII, VIII, and IX provide some variants of its implementation.
>
>
> Q2: Motivation of modeling a task as a graph (“Things to improve the paper 2”)
> A2: To summarize and represent a task, we extract and aggregate the information across multiple prototypes, which serves as one of the key steps in our solution. To achieve this goal, we propose to construct a graph between prototypes within a task and then use it to interact with meta-knowledge graph. Finally, we aggregate the information of enhanced prototypes to derive the task embedding. The graph structure is chosen as we not only need to model the prototype information but also the complex relationship among prototypes, as prototypes may be correlated to each other.
> To further verify the effectiveness of graph structure, we further conduct one more ablation:
>     - Removing the link between prototypes (ablation III in the revised paper):
>         - Results of Plain-Multi: bird 72.53±0.72% | texture 49.25±0.68% | aircraft 74.46±0.64% | fungi 57.10±0.81%
>         - Results of Art-Multi: ave. original 61.23±0.75% | ave. blur 58.43±0.76% | ave. pencil 54.76±0.72%
> The better performance of ARML than ablation model III and ablation model I (no prototype-based graph) demonstrates the effect of the relation between prototypes.
>
>
> Q3: Results of MiniImagenet and tieredImagenent (“Things to improve the paper 3”)
> A3: For MiniImagenet, the performance had already been reported. Please refer to Section D in the appendix. For tieredImagenet, we show the performance on 5-way 1-shot as follows (several MAML-based models are selected for comparison):
>     - Performance on globally shared models: MAML: 51.37 ± 1.80% | Reptile: 49.41 ± 1.82% | MetaSGD 51.48 ± 1.79%
>     - Task-specific models: MT-Net: 51.95 ± 1.83% | MUMOMAML: 51.95 ± 1.80% | HSML: 52.67 ± 1.85%
>     - Our ARML 52.91 ± 1.83%
> Since these two benchmarks do not have obvious task heterogeneity, similar to the settings in [Finn NeurIPS’18], the goal is to compare our model with other MAML-based models and report the results. We can see our ARML achieves comparable performance on these two homogeneous benchmarks but better performance on heterogeneous datasets (i.e., Plain-Multi and Art-Multi).

---

> > ### Author Response · Authors · 2019-11-12
> > **Response to Review #1, part 2/2**
> >
> >
> > Q4: More baselines (“Things to improve the paper 3”)
> > A4: We always strive to conduct fair and comprehensive experimental comparisons. Therefore, we compare the proposed approach with representative gradient-based meta-learning methods, including several task-specific gradient-based meta-learning models, which represent the state-of-the-art in the investigated research field, and two related non-parametric based meta-learning models. To make the comparisons more comprehensive, we add three more recent works as baselines, i.e., BMAML, TapNet, and VERSA. The 5-way 1-shot results on Plain-Multi and Art-Multi datasets are shown as follows (see the revised paper for 5-way 5-shot results in Table 1 and 2 and full results in Table 10):
> >     - BMAML [T Kim NeurIPS’18] (MAML-based method)
> >         - Results of Plain-Multi: bird 54.89±1.48% | texture 32.53±1.33% | aircraft 53.63±1.37% | fungi 42.50±1.33%
> >         - Results of Art-Multi: ave. original 43.66±1.36% | ave. blur 41.08±1.35% | ave. pencil 37.28±1.39%
> >     - TapNet [S Yoon ICML’19] (non-parametric method)
> >         - Results of Plain-Multi: bird 54.90±1.34% | texture 32.44±1.23% | aircraft 51.22±1.34% | fungi 42.88±1.35%
> >         - Results of Art-Multi: ave. original 42.15±1.36% | ave. blur 41.16±1.34% | ave. pencil 37.25±1.33%
> >     - VERSA [J Gordon ICLR’19] (black-box amortized method)
> >         - Results of Plain-Multi: bird 53.40±1.41% | texture 30.43±1.30% | aircraft 50.60±1.34% | fungi 40.40±1.40%
> >         - Results of Art-Multi: ave. original 43.91±1.35% | ave. blur 41.98±1.35% | ave. pencil 38.70±1.33%
> > Note that, for all baselines and ARML, we use the same base learner (4-block convolutional model) used in (Finn ICML’17). We observe that the proposed approach outperforms these three baselines as well under different experiment settings.
> >
> >
> > Q5: Semantic explanation of learned meta-knowledge graph (“Main arguments 2”)
> > A5: We agree that it is non-intuitive and hard to derive concrete interpretations of every detail of the meta-knowledge graph. But the focus of introducing such a meta-knowledge graph is to maintain the historical learned knowledge and model the complex relationship among them. The relationship gauges the relevance between different knowledge. With the meta-knowledge graph at hand, we can accomplish new tasks more effectively by paying attention to knowledge, which are highly relevant. The qualitative results in Figure 2 and 3 justify the contribution of the meta-knowledge graph and provide insights of learned relations. Nevertheless, as future work, we are excited to investigate the explainable semantic meaning in the meta-knowledge graph on this problem. Thank you for this wonderful suggestion.
> >
> >
> > References:
> > [Finn NeurIPS’18] Finn, Chelsea, Kelvin Xu, and Sergey Levine. "Probabilistic model-agnostic meta-learning." Advances in Neural Information Processing Systems. 2018.
> > [Kim NeurIPS’18] Kim, Taesup, et al. "Bayesian model-agnostic meta-learning." Advances in Neural Information Processing Systems. 2018.
> > [Yoon ICML’19] Yoon, Sung Whan, Jun Seo, and Jaekyun Moon. "TapNet: Neural Network Augmented with Task-Adaptive Projection for Few-Shot Learning." International Conference on Machine Learning. 2019.
> > [Gordon ICLR’19] Gordon, Jonathan, et al. "Meta-Learning Probabilistic Inference for Prediction." ICLR (2019).
> > [Finn ICML’17] Finn, Chelsea, Pieter Abbeel, and Sergey Levine. "Model-agnostic meta-learning for fast adaptation of deep networks." Proceedings of the 34th International Conference on Machine Learning-Volume 70. JMLR. org, 2017.

---

### Author Response · Authors · 2019-11-14
**Summary of Paper Revision**

We sincerely appreciate all the reviewers for their constructive comments to improve our paper. We have revised our paper and the major changes are:
    - Added three ablation models in Table 3 and Table 7 (Appendix F) based on the valuable comments of Reviewer 1 and 3.
    - We reported the comparison of tieredImagenet in Table 5 (Appendix D) as recommended by Reviewer 1.
    - As suggested by Reviewer 1, three more baselines (BMAML, VERSA, TapNet) are added for comparison in Table 1, 2, 10.
    - Compared the learned structure between HSML and ARML in Appendix H.1 based on our response to Reviewer 2's constructive comments.
    - Provided more cases to analyze the meta-knowledge graph in Appendix H.2 based on the suggestion of Reviewer 3.
    - Moved "Performance v.s. Vertice Numbers" part from section 5.3.2 to Appendix G.2 due to the space limitation.

We have highlighted these major changes in the revised version.

---

### Decision · Program_Chairs · 2019-12-19

**Decision:**

Accept (Poster)

**Comment:**

This paper proposes to deal with task heterogeneity in meta-learning by extracting cross-task relations and constructing a meta-knowledge graph, which can then quickly adapt to new tasks. The authors present a comprehensive set of experiments, which show consistent performance gains over baseline methods, on a 2D regression task and a series of few-shot classification tasks. They further conducted some ablation studies and additional analyses/visualization to aid interpretation.

Two of the reviewers were very positive, indicating that they found the paper well-written, motivated, novel, and thorough, assessments that I also share. The authors were very responsive to reviewer comments and implemented all actionable revisions, as far as I can see. The paper looks to be in great shape. I’m therefore recommending acceptance.